# LANGUAGE REPOSITORY FOR LONG VIDEO UNDERSTANDING

## ABSTRACT

Language has become a prominent modality in computer vision with the rise of LLMs. Despite supporting long context-lengths, their effectiveness in handling long-term information gradually declines with input length. This becomes critical, especially in applications such as long-form video understanding. In this paper, we introduce a Language Repository (`LangRepo`) for LLMs, that maintains concise and structured information as an interpretable (*i.e.*, all-textual) representation. Our repository is updated iteratively based on multi-scale video chunks. We introduce write and read operations that focus on pruning redundancies in text, and extracting information at various temporal scales. The proposed framework is evaluated on zero-shot visual question-answering benchmarks including EgoSchema, NExT-QA, IntentQA and NExT-GQA, showing state-of-the-art performance at its scale. Our code will be made publicly available.

## 1 INTRODUCTION

Video data is central to learning systems that can interact and reason about the world. Yet, they also associate with significant challenges such as increased compute requirements and redundant information, to name a few. This is especially critical in long-form videos. Even so, recent literature on video understanding have progressed so far, enabling reasoning capabilities in hours-long video streams (Team et al., 2023; Islam et al., 2024), in contrast to very-limited temporal spans (*e.g.* seconds or minutes) just a few years ago. Such progress is intriguing considering how complex the semantics become when temporal span is increased (Sigurdsson et al., 2016; Yeung et al., 2018). Work on efficient spatio-temporal attention mechanisms (Arnab et al., 2021; Bertasius et al., 2021), memory management (Wu et al., 2022; Ryoo et al., 2023), and large-language-models (LLMs) (Wang et al., 2022a; Yu et al., 2024; Team et al., 2023) have been key ingredients for such improvements.

LLMs, or more-specifically, vision-large-language-models (VLLMs) have been outperforming pure vision models in recent years in all facets, including image-based reasoning (Liu et al., 2024; Zheng et al., 2024; Li et al., 2023b), grounding (Lai et al., 2023; Rasheed et al., 2023), video understanding (Wang et al., 2022a; Ye et al., 2023b; Yu et al., 2024), and even robotics (Zeng et al., 2022; Ahn et al., 2022; Liang et al., 2023; Li et al., 2024b). The sheer model scale and the vast pretraining data have enabled such frameworks to capture world knowledge and semantics, beyond what is possible with visual data only. Besides, the ability to process long context-lengths is also key, as it helps modeling long-term dependencies that are crucial for more-complex reasoning and interactions. However, recent studies show that despite the availability of such context-lengths, the effectiveness of models declines with longer input sequences (Levy et al., 2024). This promotes the search for alternate representations that can compress input language data without losing meaningful information, essentially managing the context utilization of LLMs.

Moreover, the use of text (*i.e.*, language) in modeling has shown numerous benefits such as rich semantics (Wang et al., 2022b; Menon & Vondrick, 2022; Kahatapitiya et al., 2023), ease of information sharing between different specialized-models (Zeng et al., 2022) or modalities (Liu et al., 2024; Girdhar et al., 2023), and interpretability (Zhao et al., 2023a; Singh et al., 2024). Among such, interpretability has a huge societal impact in the age of LLMs, to manage adversities such as bias (Liang et al., 2021; Ferrara, 2023) and hallucinations (Zhang et al., 2023b; Dhuliawala et al., 2023). Simply put, it enables human observers to understand and monitor what really happens within mod-

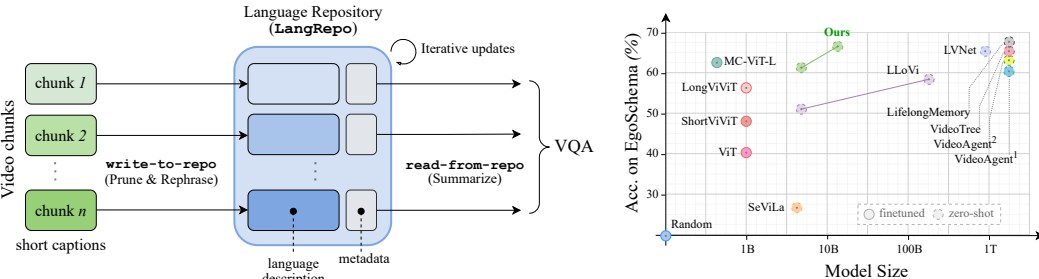

Figure 1: **Overview of our Language Repository (`LangRepo`):** We propose an all-textual repository of visual information that updates iteratively, creating a multi-scale and interpretable representation. It extracts information from captions corresponding to video chunks, generated by a VLLM. In `write-to-repo`, we prune and rephrase input descriptions, creating concise entries in the repository. In `read-from-repo`, such language descriptions (together with any optional metadata, *e.g.*, timestamps) at multiple semantic-scales are summarized to generate outputs suited for video VQA. Here, rephrase and summarize are LLM-calls. We also compare `LangRepo` against state-of-the-art methods, showing strong performance at its scale.

els. Hence, interpretable representations have also been of interest to the community, in place of latent representations (Wu et al., 2022; Ryoo et al., 2023).

Motivated by the above, we introduce Language Repository (`LangRepo`), an interpretable representation for LLMs that updates iteratively. It consumes input captions corresponding to video chunks , as shown in Fig. 1 (left). As `LangRepo` is all-textual, we rely on text-based operations to write and read information. The write operation (`write-to-repo`) prunes redundant text, creating concise descriptions that keep the context-utilization of LLMs in-check. Its iterative application with increasingly-longer chunks enables it to learn high-level semantics (*e.g.* long temporal dependencies). The read operation (`read-from-repo`) extracts such stored language information at various temporal scales, together with other optional metadata within the repository entries (*e.g.* timestamps). Altogether, our proposed framework is applied to long-term video reasoning tasks including visual question-answering (VQA) on EgoSchema (Mangalam et al., 2024), NExT-QA (Xiao et al., 2021) and IntentQA (Li et al., 2023a), and visually-grounded VQA on NExT-GQA (Xiao et al., 2023a), showing strong performance at its scale, as given in Fig. 1 (right). Finally, we ablate our design decisions, providing insights on key components.

## 2 RELATED WORK

**Long-video understanding:** Video models have progressed over the years, going from primitive recognition tasks (Soomro et al., 2012; Kuehne et al., 2011) to complex and fine-grained reasoning tasks (Sigurdsson et al., 2016; Yeung et al., 2018; Xiao et al., 2021; Grauman et al., 2022; Mangalam et al., 2024) over long horizons. Both convolutional baselines (Carreira & Zisserman, 2017; Feichtenhofer et al., 2019; Feichtenhofer, 2020) and transformer architectures (Arnab et al., 2021; Bertasius et al., 2021; Nagrani et al., 2021) have explored research directions such as multi-scale representations (Feichtenhofer et al., 2019; Fan et al., 2021; Liu et al., 2022), efficiency concerns associated with heavy spatio-temporal computations (Duke et al., 2021; Li et al., 2019), and handling redundant information within video inputs (Chen et al., 2018; Kahatapitiya & Ryoo, 2021). More recently, long-video understanding has made a leap forward thanks to benchmark datasets (Grauman et al., 2022; Mangalam et al., 2024; Xiao et al., 2021) and model improvements (Yu et al., 2024; Zhang et al., 2023a; Papalampidi et al., 2023), validating the importance of modeling complex interactions that happen over long periods of time. Still, the sub-par performance of SOTA models on such benchmarks suggests the room for improvement.

**Long-context models:** Even before the age of LLMs, models based on convolutions (Wang et al., 2018; Piergiovanni & Ryoo, 2018; 2019; Kahatapitiya & Ryoo, 2021), recurrent blocks (Greff et al., 2016; Chung et al., 2014; Hutchins et al., 2022) or transformers (Wu et al., 2022; Ryoo et al., 2023; Chen et al., 2021) have exploited long-term dependencies, especially in the context of video understanding (Wang et al., 2018; Wu et al., 2022) and robotics (Chen et al., 2021; Shang et al., 2022).

With the rise of LLMs, scaling laws have revealed the importance of longer contexts even more (Team et al., 2023; Reid et al., 2024), and, thanks to the breakthroughs such as sparse processing (Shazeer et al., 2017; Fedus et al., 2022), caching (Ge et al., 2023; Kwon et al., 2023; Khandelwal et al., 2018), model-sharding (Zhao et al., 2023b; Chowdhery et al., 2023; Lepikhin et al., 2020), and efficient attention (Dao et al., 2022; Lefaudeux et al., 2022), such long-context LLMs have become a reality. Even with very large context lengths, maintaining the effectiveness of reasoning over longer inputs is challenging (Levy et al., 2024; Xiong et al., 2023; Shi et al., 2023). This motivates us to think about concise representations that can better-utilize LLM context.

**Compressing representations:** When handling heavy inputs, deep learning models have relied on compressed representations. It may come in the form of pruning (Ryoo et al., 2021; Bolya et al., 2022), latent memory (Ryoo et al., 2023; Graves et al., 2014; Wu et al., 2022), or external feature banks (Wu et al., 2019), to name a few. Despite the intuitive novelties and efficiency gains of such techniques, it is challenging to realize which information gets preserved, and how semantically-meaningful they are post-compression. An interpretable representation that supports compression, if available, may shed light on such details.

**Language as an interpretable modality:** More-recently, language has emerged as a dominant modality in computer vision due to its strong generalization capabilities (Radford et al., 2021; Jia et al., 2021). It can also act as a bridge between various domain-specific models (Zeng et al., 2022), other modalities (Liu et al., 2024; Girdhar et al., 2023), and even human instructions (Surís et al., 2023; Gupta & Kembhavi, 2023), showing intriguing applications in domains such as chat agents (*e.g.* ChatGPT, Bard) and robotics (Ahn et al., 2022; Liang et al., 2023). Since language is interpretable, it enables humans to interact with models naturally and make sense of model predictions.

Motivated by the above, we introduce an interpretable language representation that can (1) prune redundant information, and (2) extract multi-scale (or, high-level) semantics, enabling better context-utilization within LLMs. We rely on open-source LLMs without additional video pretraining, yet showing a strong performance compared to concurrent work based on much-larger proprietary models (Park et al., 2024; Wang et al., 2024b; Fan et al., 2024; Wang et al., 2024e;d; Kim et al., 2024) or video-pertained multi-modal models (Wang et al., 2024a; Li et al., 2024a; Wang et al., 2024c).

## 3 OBSERVATIONS ON LONG-RANGE INPUTS

In this section, we investigate how LLMs perform with increasing inputs lengths (*i.e.*, #tokens). Recent LLMs with very-large context lengths such as Gemini-Pro-1.5 (Team et al., 2023) (1M tokens) or Claude-2.1 (200k tokens), can support extremely long input sequences. Yet, when feeding longer inputs, the reasoning capabilities (especially, long-term reasoning) of such models diminish. This behavior is also observed in concurrent work (Levy et al., 2024), and evident in benchmark results of state-of-the-art models (Ye et al., 2023b; Yu et al., 2024) (*i.e.*, better performance with shorter inputs, or fewer video frames). To better investigate this in our setup, we evaluate VQA performance on standard long-term video understanding benchmarks while varying the input length (see Table 1). We consider frame/short-clip captions extracted using a VLLM at a baseline framerate (1×) as inputs (in-

Table 1: **Observations on increasing input length:** We evaluate the VQA performance of an LLM (Jiang et al., 2023) at different input lengths, on multiple long-video benchmarks (Mangalam et al., 2024; Xiao et al., 2021; Li et al., 2023a). Even with a sufficient context length, the effectiveness of predictions decreases with longer input. Here, 1× corresponds to captions generated at a standard frame-rate (and, 0.5×/2× corresponds to a compression/expansion by a factor of 2).

| Dataset | Captions per-video | | |
|---|---|---|---|
| | 0.5× | 1× | 2× |
| EgoSchema | 49.8 | 48.8 | 46.8 |
| NExT-QA | 48.2 | 48.2 | 46.9 |
| IntentQA | 47.1 | 46.9 | 45.2 |

troduced in (Zhang et al., 2023a)). We either subsample (0.5×) or replicate (2×) the captions, decreasing/increasing the input lengths of a question-answering LLM, namely, Mistral-7B (Jiang et al., 2023) with 8k (or, theoretical 128k) context length. All inputs fit within the context, without any overflow. The observation from this study is consistent: even though the context length of the LLM is sufficient to process given inputs, the effectiveness of its predictions (shown by VQA performance) drops with longer inputs. This motivates us to introduce a concise language representation that preserves important details of long-range inputs, while pruning any redundant information.

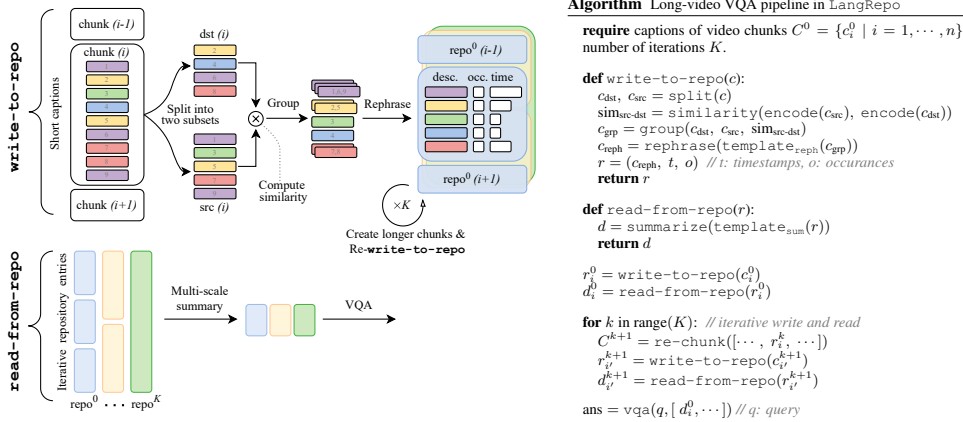

Figure 2: **Detailed view of our Language Repository (`LangRepo`):** Here we present the write and read operations within `LangRepo`. Given short-captions corresponding to video chunks, `write-to-repo` first prunes redundant captions within each chunk. The same process is iteratively applied on increasingly longer (or, higher-level) chunks— that are already within the repository— to generate multi-scale repository entries. Pruning consists of two stages: (1) grouping most similar captions based on embedding (*e.g.* CLIP (Radford et al., 2021)) similarities between two subsets, and (2) rephrasing grouped captions with an LLM-call. The resulting `LangRepo` will include rephrased-captions and any optional metadata (*e.g.* #occurrences, timestamps). Next, `read-from-repo` generates concise descriptions for different semantic levels by summarizing the multi-scale language representation, which is also an LLM-call.

## 4 LANGUAGE REPOSITORY

We present a Language Repository (`LangRepo`) that iteratively updates with multi-scale descriptions from video chunks. In contrast to external feature banks (Wu et al., 2019) or learnable latent memory representations (Wu et al., 2022; Ryoo et al., 2023; Balažević et al., 2024), our proposal has a few key advantages: (1) it requires no training (*i.e.*, zero-shot), and (2) it is compatible with both LLM-based processing and human interpretation, as it is fully-textual, *i.e.*, it exists in language-space instead of a latent-space. `LangRepo` consists of two main operations: (1) information writing (`write-to-repo`), which prunes redundancies and iteratively updates language descriptions based on increasingly-longer video chunks, and (2) information reading (`read-from-repo`), which extracts preserved descriptions (with any optional metadata) in multiple temporal scales. We show a detailed view of these operations in Fig. 2, and further elaborate in the following subsections.

Consider a long video that is split in to $n$ non-overlapping chunks, denoted as $V = \{v_i \mid i = 1, \cdots, n\}$. Assume that we already have frame or short-clip captions extracted by a VLLM (*e.g.* LLaVA (Liu et al., 2024)) corresponding to such chunks, denoted by $C^0 = \{c_i^0 \mid i = 1, \cdots, n\}$. Here, each chunk may consist of $p$ such captions as in $c_i^0 = \{c_{ij}^0 \mid j = 1, \cdots, p\}$. Altogether, $V$ is represented by $n \times p$ captions which we consider as inputs to our framework.

### 4.1 WRITING TO REPOSITORY

We intend to create a concise, all-textual representation with multiple scales (or, semantic-levels) of information. Hence, our writing operation is text-based, and applied iteratively on different scales of input. In the first iteration, it consumes low-level details in each chunk $i$, in the form of captions $c_i^0$, generating initial entries to the repository $\text{repo}^0(i)$, or $r_i^0$.

$$r_i^0 = \texttt{write-to-repo}(c_i^0) . \tag{1}$$

In each subsequent iteration $k + 1$, previous repo entries of iteration $k$ are re-combined into longer chunks and processed in the same way, generating information for higher semantic-levels.

$$[c_1^{k+1}, \cdots, c_m^{k+1}] = \texttt{re-chunk}([r_1^k, \cdots, r_n^k]) , \tag{2}$$

$$r_{i'}^{k+1} = \texttt{write-to-repo}(c_{i'}^{k+1}) . \tag{3}$$

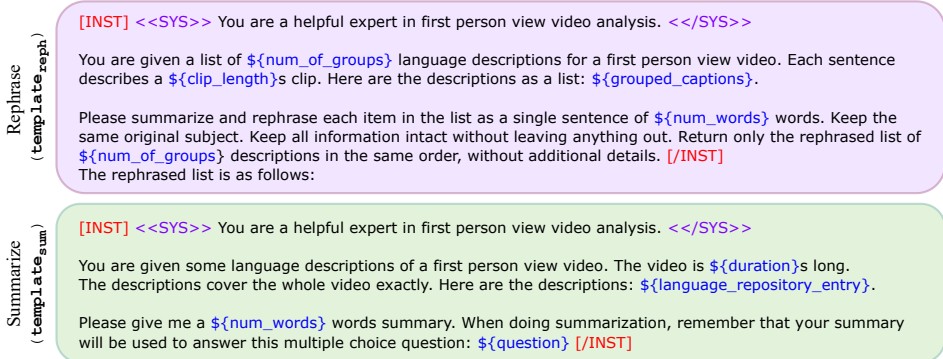

Figure 3: **LLM prompt templates in `LangRepo`:** Here, we show the zero-shot prompt templates used for rephrasing ($\texttt{template}_{\text{reph}}$) and summarizing ($\texttt{template}_{\text{sum}}$) operations. Rephrase prompt needs a list of grouped captions as input, while its output adheres to more-strict requirements (*e.g.* same order, same number of list items) needed for correct parsing. Summarize prompt takes in each repository entry and generates a more-flexible (*i.e.*, open-ended) output, while optionally conditioning on the question.

Here, $\texttt{re-chunk}(\cdot)$ denotes the creation of longer (and, fewer, *i.e.*, $m < n$) chunks within the repository. More specifically, we simply concatenate (denoted by $[\cdot]$) all entries from previous iteration, and split them again into fewer number of chunks (hence, longer chunk size). Note that $i'$ in the above equation is not the same as the previous chunk indexing $i$, as we may have different (usually, fewer) number of chunks in each subsequent iteration. Each write operation involves two stages: (1) Grouping redundant text, and (2) Rephrasing, which are detailed below.

**Grouping redundant text:** Given textual descriptions of a video chunk (*i.e.*, captions in the first write iteration, or previous repo descriptions in subsequent iterations), we plan to identify most-similar ones and merge them as a single description. Without loss of generality, let us consider the first write iteration, for which the input is in the form of $c_i^0 = \{c_{ij}^0 \mid j = 1, \cdots, p\}$. Inspired by (Bolya et al., 2022), we first split the captions of each chunk into two sets, namely, source (*src*) captions $c_{\text{src},i}^0$ and destination (*dst*) captions $c_{\text{dst},i}^0$. Let us drop the chunk index ($i$) and iteration index (0) for brevity. Here, dst captions $c_{\text{dst}}$ are sampled uniformly distributed across the temporal span of a chunk, while all the rest are considered as src captions $c_{\text{src}}$ (see Fig. 2 top-left).

$$c_{\text{dst}}, \; c_{\text{src}} = \texttt{split}(c) \,. \tag{4}$$

Here, we usually have fewer dst captions (*i.e.*, $|c_{\text{dst}}| < |c_{\text{src}}|$). Next, we embed all captions using a text-encoder (*e.g.* CLIP (Radford et al., 2021)), and compute the cosine similarity of each pair between src-dst sets to find most-similar matches.

$$\text{sim}_{\text{src-dst}} = \texttt{similarity}(\texttt{encode}(c_{\text{src}}), \, \texttt{encode}(c_{\text{dst}})) \,. \tag{5}$$

Based on the similarity matrix above ($\text{sim}_{\text{src-dst}}$), we then prune the highest $x\%$ similarities by grouping such source captions with their corresponding destination matches, forming a set of grouped descriptions $c_{\text{grp}}$ for the given chunk. Refer to the color-coded captions after 'Group' in Fig. 2.

$$c_{\text{grp}} = \texttt{group}(c_{\text{dst}}, \; c_{\text{src}}, \; \text{sim}_{\text{src-dst}}) \,. \tag{6}$$

Here, an additional hyperparameter (*i.e.*, $x$) decides the grouping ratio. Finally, such grouped descriptions go through a rephrasing operation prior to entering the repository.

**Rephrasing:** Grouped captions $c_{\text{grp}}$ of each chunk are rephrased via an LLM-call. This allows redundant information within each group to be dropped, while generating a concise and coherent description. We first form a list of grouped captions, where each list item corresponds to a single group (*i.e.*, a dst caption and any one or more src captions matched to it), and feed it to the LLM, wrapped in a rephrasing-template ($\texttt{template}_{\text{reph}}$) as shown in Fig. 3 (top-left).

$$c_{\text{reph}} = \texttt{rephrase}(\texttt{template}_{\text{reph}}(c_{\text{grp}})) \,. \tag{7}$$

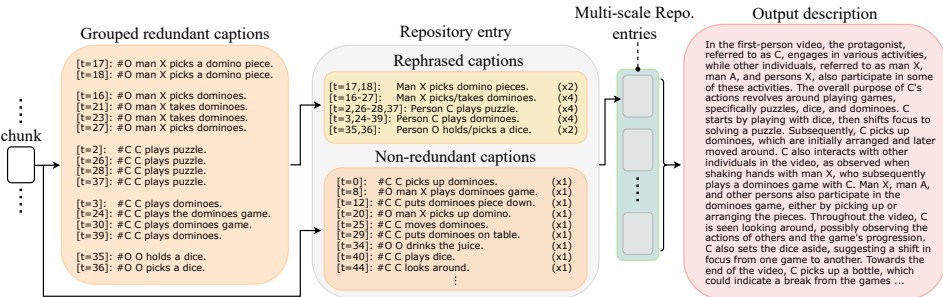

Figure 4: **A qualitative example of a `LangRepo` entry:** Given a video chunk, redundant captions are first grouped together during pruning operation. During rephrasing, such groups are more-concisely written to the repository, along with additional metadata. Other non-redundant captions are written directly. This process is continued iteratively with increasingly-longer chunks, creating multi-scale repository entries (refer Fig. A.1 for a more-detailed view). Finally, such descriptions from various temporal scales are read to generate the output.

Here, the LLM output ($c_{\text{reph}}$) is restricted to be a list in the same order with the same number of items, where each item is a single concise sentence. Finally, such rephrased descriptions together with other metadata such as timestamps ($t$) and number of occurrences ($o$) are written in the repository.

$$r = \{(c_{\text{reph},j},\ t_j,\ o_j) \mid j = 1,\ \cdots,\ p'\}\,. \tag{8}$$

Note that here $p' < p$ as we have grouped and rephrased a pre-defined ratio (*e.g.* 50%) of most-similar captions. Alongside each description in a repository entry, $t$ maintains a list of timestamps corresponding to its founding captions, whereas the occurrences counter ($o$) keeps track of the number of captions grouped together. A qualitative example of a repository entry is given in Fig. 4.

In subsequent iterations, the same operations apply when writing multi-scale entries. The only difference is the change in input, which now constitutes of previous repo entries re-combined into high-level chunks (*i.e.*, $c^0 \rightarrow c^k$). Each new iteration generates information corresponding to a higher semantic-level (*i.e.*, going from short-range to long-range dependencies), forming our multi-scale language representation.

## 4.2 READING FROM REPOSITORY

As we make a single VQA prediction for a given long video— instead of making predictions every chunk— our read operation (`read-from-repo`) is applied after fully-forming each scale of multi-scale repository (*i.e.*, after writing all chunks). The repo entries from $K$ scales can be denoted as $\{r^k \mid k = 0, \cdots, K\}$ where each scale ($r^k$) may consist of multiple entries $\{\cdots, r^k_{i-1}, r^k_i, r^k_{i+1}, \cdots\}$. When reading, we generate summaries for each entry in the repo separately, allowing it to focus on varying temporal spans. More specifically, each entry goes through a summarizing-template (`template_sum`) as shown in Fig. 3 (bottom), and the resulting prompt is fed to the LLM.

$$d^k_i = \texttt{read-from-repo}(r^k_i) = \texttt{summarize}(\texttt{template}_{\text{sum}}(r^k_i))\,. \tag{9}$$

Here, $d^k_i$ corresponds to the output description of each entry $i$ in the repository, at the respective scale $k$. Optionally, we can make use of additional metadata such as timestamps and #occurrences, by prompting the read operation with descriptions of repo entries formatted as "`[timestamps] description (×#occurrences)`" (see Fig. 4). Finally, we concatenate all output descriptions and prompt the LLM again to generate the answer prediction.

$$\texttt{ans} = \texttt{vqa}([\cdots, d^k_i, \cdots])\,. \tag{10}$$

## 5 EXPERIMENTS

In our experiments, we rely on captions pre-extracted using VLLMs, as given in (Zhang et al., 2023a). As for the LLM, we use either Mistral-7B (Jiang et al., 2023) (w/ 7B parameters) or Mixtral-

Table 2: **Results on EgoSchema (Mangalam et al., 2024):** We present comparisons with state-of-the-art models on EgoSchema subset (500-videos) and fullset (5000-videos). We focus on the zero-shot setting. `LangRepo` shows a strong performance at its scale.

| Model | Video Pretrain | Params | Subset (%) | Fullset (%) |
|---|---|---|---|---|
| *finetuned* | | | | |
| MC-ViT-L (Balažević et al., 2024) | ✓ | 424M | 62.6 | 44.4 |
| ImageViT (Papalampidi et al., 2023) | ✓ | 1B | 40.8 | 30.9 |
| ShortViViT (Papalampidi et al., 2023) | ✓ | 1B | 47.9 | 31.0 |
| LongViViT (Papalampidi et al., 2023) | ✓ | 1B | 56.8 | 33.3 |
| *zero-shot (with proprietary LLMs)* | | | | |
| Vamos (Wang et al., 2023) | ✓ | 175B | - | 41.2 |
| Vamos (Wang et al., 2023) | ✓ | 1.8T | - | 48.3 |
| LLoVi (Zhang et al., 2023a) | ✗ | 175B | 57.6 | 50.3 |
| ProViQ (Choudhury et al., 2023) | ✗ | 175B | - | 57.1 |
| MoReVQA (Min et al., 2024) | ✗ | 340B | - | 51.7 |
| LVNet (Park et al., 2024) | ✗ | <1.8T | 68.2 | 61.1 |
| VideoAgent (Wang et al., 2024b) | ✗ | 1.8T | 60.2 | 54.1 |
| VideoAgent (Fan et al., 2024) | ✗ | 1.8T | 62.8 | - |
| IG-VLM (Kim et al., 2024) | ✗ | 1.8T | - | 59.8 |
| VideoTree (Wang et al., 2024e) | ✗ | 1.8T | 66.2 | 61.1 |
| LifelongMemory (Wang et al., 2024d) | ✗ | 1.8T | 68.0 | 62.1 |
| *zero-shot (with open-source LLMs)* | | | | |
| VIOLET(Fu et al., 2023) | ✓ | 198M | - | 19.9 |
| InternVideo (Wang et al., 2022a) | ✓ | 478M | - | 32.1 |
| FrozenBiLM (Yang et al., 2022) | ✓ | 890M | - | 26.9 |
| SeViLA (Yu et al., 2024) | ✓ | 4B | 25.7 | 22.7 |
| Tarsier (Wang et al., 2024a) | ✓ | 7B | 56.0 | 49.9 |
| VideoChat2 (Li et al., 2024a) | ✓ | 7B | 63.6 | 54.4 |
| VideoLLaMA 2 (Cheng et al., 2024) | ✓ | 12B | - | 53.3 |
| Vamos (Wang et al., 2023) | ✓ | 13B | - | 36.7 |
| InternVideo2 (Wang et al., 2024c) | ✓ | 13B | - | 60.2 |
| Tarsier (Wang et al., 2024a) | ✓ | 34B | 68.6 | 61.7 |
| mPLUG-Owl (Ye et al., 2023b) | ✗ | 7B | - | 31.1 |
| Mistral (Jiang et al., 2023) | ✗ | 7B | 48.8 | - |
| LLoVi (Zhang et al., 2023a) | ✗ | 7B | 50.8 | 33.5 |
| LangRepo (ours) | ✗ | 7B | 60.8 | 38.9 |
| LangRepo (ours) | ✗ | 12B | 66.2 | 41.2 |

8×7B (Jiang et al., 2024) (w/ 12B active parameters) by default. As the text encoder in similarity-based pruning, we use CLIP-L/14 (Radford et al., 2021). Note that all the models used in our framework are open-source and within a reasonable model-scale, making our work accessible even in academic settings. We do zero-shot inference on all datasets without any finetuning, evaluating the performance on long-form video VQA benchmarks.

For evaluations, we consider four challenging long-video VQA benchmarks in our evaluations. EgoSchema (Mangalam et al., 2024) derived from Ego4D (Grauman et al., 2022), consists of 3-minute long clips, each with a question and 5 answer-choices. Its public validation subset consists of 500 videos, whereas the held-out fullset has 5K videos. NExT-QA (Xiao et al., 2021) contains videos up to 2 minutes long (at an average of 44 seconds), annotated with 52k open-ended questions and 48k close-ended questions (*i.e.*, multiple-choice with 5 answer options). The questions are further classified into temporal, causal, or descriptive categories, to evaluate different reasoning capabilities of models. We consider zero-shot evaluation on the validation set. IntentQA (Li et al., 2023a) is based on the same NExT-QA videos, yet focuses more on intent-related questions (*e.g.* why?, how? or before/after) with a total of 16k multiple-choice questions on 4.3k videos. Here, we consider zero-shot setting on the test set. NExT-GQA (Xiao et al., 2023a) is a visually-grounded VQA dataset with 10.5K temporal grounding annotations, where we consider zero-shot inference similar to (Zhang et al., 2023a), on the test split.

## 5.1 MAIN RESULTS

**EgoSchema:** In Table 2, we present the VQA performance of `LangRepo` on standard EgoSchema (Mangalam et al., 2024) splits, comparing with other state-of-the-art frameworks. Here, we focus on zero-shot evaluation, yet also report finetuned setting (*i.e.*, any downstream-data-specific training) for completeness. We consider Mistral-7B (Jiang et al., 2023) and Mixtral-8×7B (Jiang et al., 2024) as the choice of LLMs in our setup, both with reasonable model scales (7B and 12B active parameters, respectively). We de-emphasize the comparisons with models having significantly-higher #parameters (*e.g.* 175B GPT-3.5, or 1.8T GPT-4 variants), and multi-modal LLMs that use video-

Table 3: **Results on NExT-QA (Xiao et al., 2021):** We compare `LangRepo` against state-of-the-art methods on NExT-QA validation set, highlighting standard splits: causal, temporal and descriptive. We focus on the zero-shot setting. Our method shows strong performance at its scale.

| Model | Video Pretrain | Params | Causal (%) | Temporal (%) | Descriptive (%) | All (%) |
|---|---|---|---|---|---|---|
| *finetuned* | | | | | | |
| CoVGT (Xiao et al., 2023b) | ✓ | 149M | 58.8 | 57.4 | 69.3 | 60.0 |
| SeViT$_{FiD}$ (Kim et al., 2023) | ✓ | 215M | - | - | - | 60.6 |
| HiTeA (Ye et al., 2023a) | ✓ | 297M | 62.4 | 58.3 | 75.6 | 63.1 |
| MC-ViT-L (Balažević et al., 2024) | ✓ | 424M | - | - | - | 65.0 |
| InternVideo (Wang et al., 2022a) | ✓ | 478M | 62.5 | 58.5 | 75.8 | 63.2 |
| BLIP-2 (Li et al., 2023b) | ✓ | 4B | 70.1 | 65.2 | 80.1 | 70.1 |
| SeViLA (Yu et al., 2024) | ✓ | 4B | 74.2 | 69.4 | 81.3 | 73.8 |
| LLama-VQA (Ko et al., 2023) | ✓ | 7B | 72.7 | 69.2 | 75.8 | 72.0 |
| Vamos (Wang et al., 2023) | ✓ | 7B | 72.6 | 69.6 | 78.0 | 72.5 |
| *zero-shot (with proprietary LLMs)* | | | | | | |
| ViperGPT (Surís et al., 2023) | ✗ | 175B | - | - | - | 60.0 |
| ProViQ (Choudhury et al., 2023) | ✗ | 175B | - | - | - | 64.6 |
| MoReVQA (Min et al., 2024) | ✗ | 340B | 70.2 | 64.6 | - | 69.2 |
| LVNet (Park et al., 2024) | ✗ | <1.8T | 75.0 | 65.5 | 81.5 | 72.9 |
| IG-VLM (Kim et al., 2024) | ✗ | 1.8T | 69.8 | 63.6 | 74.7 | 68.6 |
| LLoVi (Zhang et al., 2023a) | ✗ | 1.8T | 69.5 | 61.0 | 75.6 | 67.7 |
| TraveLER (Shang et al., 2024) | ✗ | 1.8T | 70.0 | 60.5 | 78.2 | 68.2 |
| VideoAgent (Wang et al., 2024b) | ✗ | 1.8T | 72.7 | 64.5 | 81.1 | 71.3 |
| VideoTree (Wang et al., 2024e) | ✗ | 1.8T | 75.2 | 67.0 | 81.3 | 73.5 |
| *zero-shot (with open-source LLMs)* | | | | | | |
| VFC (Momeni et al., 2023) | ✓ | 164M | 45.4 | 51.6 | 64.1 | 51.5 |
| InternVideo (Wang et al., 2022a) | ✓ | 478M | 43.4 | 48.0 | 65.1 | 49.1 |
| SeViLA (Yu et al., 2024) | ✓ | 4B | 61.3 | 61.5 | 75.6 | 63.6 |
| Tarsier (Wang et al., 2024a) | ✓ | 7B | - | - | - | 71.6 |
| Tarsier (Wang et al., 2024a) | ✓ | 34B | - | - | - | 79.2 |
| Mistral (Jiang et al., 2023) | ✗ | 7B | 51.0 | 48.1 | 57.4 | 51.1 |
| LLoVi (Zhang et al., 2023a) | ✗ | 7B | 55.6 | 47.9 | 63.2 | 54.3 |
| LLoVi (Zhang et al., 2023a) | ✗ | 12B | 60.2 | 51.2 | 66.0 | 58.2 |
| `LangRepo` (ours) | ✗ | 7B | 57.8 | 45.7 | 61.9 | 54.6 |
| `LangRepo` (ours) | ✗ | 12B | 64.4 | 51.4 | 69.1 | 60.9 |

caption pretraining. `LangRepo` shows significantly-better performance compared to other methods at a similar scale, validating its effectiveness. We achieve $+7.8\%$ on fullset over mPLUG-Owl (Ye et al., 2023b), $+12.0\%$ on subset over pure Mistral LLM baseline (Jiang et al., 2023), $+10.0\%$ on subset and $+5.4\%$ on fullset over LLoVi (7B) (Zhang et al., 2023a) (w/ Mistral (Jiang et al., 2023)), $+4.5\%$ on fullset over Vamos (Wang et al., 2023) (w/ Llama2 (Touvron et al., 2023)), and $+4.8\%$ on subset over Tarsier (7B) (Wang et al., 2024a).

**NExT-QA:** In Table 3, we report the performance of `LangRepo` on standard NExT-QA (Xiao et al., 2021) validation splits (Causal, Temporal and Descriptive) and the full validation set. On zero-shot evaluation, our framework outperforms other methods consistently. Compared to smaller models, we gain $+11.8\%$ over InternVideo (Wang et al., 2022a) and $+9.4\%$ over VFC (Momeni et al., 2023). Compared to models of similar scale, we gain $+3.5\%$ over baseline Mistral LLM (Jiang et al., 2023) and $+2.7\%$ over LLoVi (12B) (Zhang et al., 2023a). We de-emphasize the comparisons with much-larger models, and multi-modal LLMs pretrained with video captions (whereas we rely on LLaVA-1.5 (Liu et al., 2023) captions that has not seen any video pretraining). Finally, we observe that `LangRepo` outperforms competition on semantic splits showing the generalization of our language representation.

**IntentQA:** In Table 4, we evaluate our zero-shot framework against other state-of-the-art models on IntentQA (Li et al., 2023a) test splits (Why?, How? and Before/After) and the full test set. `LangRepo` outperform comparable models with similar scale consistently, showing gains of $+3.4\%$ over baseline Mistral LLM (Jiang et al., 2023) and $+2.5\%$ over LLoVi (12B) (Zhang et al., 2023a). Again, we de-emphasize significantly larger models and those pretrained with video-captions.

**NExT-GQA:** In Table 5, we compare the performance of `LangRepo` with state-of-the-art models on NExT-GQA (Xiao et al., 2023a). We follow the same grounding setup as in Zhang et al. (2023a). Our method achieves a strong performance at its scale, outperforming baseline Mistral LLM (Jiang et al., 2023) by $+2.0\%$ and LLoVi (12B) (Zhang et al., 2023a) by $+0.9\%$ on Acc@GQA metric. Despite being zero-shot, it is also competitive with weakly-supervised baselines. Here, we de-emphasize significantly-larger models and those pretrained with video-captions.

Table 4: **Results on IntentQA ([Li et al., 2023a](#)):** We compare `LangRepo` against state-of-the-art methods on IntentQA test set, highlighting standard splits: why?, how? and before/after. We focus on the zero-shot setting. Our method shows strong performance at its scale.

| Model | Video Pretrain | Params | Why? (%) | How? (%) | Before/After (%) | All (%) |
|---|---|---|---|---|---|---|
| *finetuned* | | | | | | |
| HQGA (Xiao et al., 2022a) | ✓ | 46M | 48.2 | 54.3 | 41.7 | 47.7 |
| VGT (Xiao et al., 2022b) | ✓ | 511M | 51.4 | 56.0 | 47.6 | 51.3 |
| Vamos (Wang et al., 2023) | ✓ | 7B | 69.5 | 70.2 | 65.0 | 68.5 |
| BlindGPT (Ouyang et al., 2022) | ✓ | 175B | 52.2 | 61.3 | 43.4 | 51.6 |
| CaVIR (Li et al., 2023a) | ✓ | 175B | 58.4 | 65.5 | 50.5 | 57.6 |
| *zero-shot (with proprietary LLMs)* | | | | | | |
| LVNet (Park et al., 2024) | ✗ | <1.8T | 75.0 | 74.4 | 62.1 | 71.7 |
| LLoVi (Zhang et al., 2023a) | ✗ | 1.8T | 68.4 | 67.4 | 51.1 | 64.0 |
| IG-VLM (Kim et al., 2024) | ✗ | 1.8T | - | - | - | 64.2 |
| VideoTree (Wang et al., 2024e) | ✗ | 1.8T | - | - | - | 66.9 |
| *zero-shot (with open-source LLMs)* | | | | | | |
| SeViLA (Yu et al., 2024) | ✓ | 4B | - | - | - | 60.9 |
| Mistral(Jiang et al., 2023) | ✗ | 7B | 52.7 | 55.4 | 41.5 | 50.4 |
| LLoVi (Zhang et al., 2023a) | ✗ | 7B | 57.9 | 55.4 | 42.3 | 53.6 |
| LLoVi (Zhang et al., 2023a) | ✗ | 12B | 59.7 | 62.7 | 45.1 | 56.6 |
| `LangRepo` (ours) | ✗ | 7B | 56.9 | 60.2 | 42.1 | 53.8 |
| `LangRepo` (ours) | ✗ | 12B | 62.8 | 62.4 | 47.8 | 59.1 |

Table 5: **Results on NExT-GQA ([Xiao et al., 2023a](#)):** We compare `LangRepo` against state-of-the-art methods on NExT-GQA test set. We focus on the zero-shot setting. Our method shows strong performance at its scale.

| Model | Video Pretrain | Params | mIoP | IoP@0.5 | mIoU | IoU@0.5 | Acc@GQA |
|---|---|---|---|---|---|---|---|
| *weakly-supervised* | | | | | | | |
| IGV (Li et al., 2022) | ✓ | 110M | 21.4 | 18.9 | 14.0 | 9.6 | 10.2 |
| Temp[CLIP] (Radford et al., 2021; Xiao et al., 2023a) | ✓ | 130M | 25.7 | 25.5 | 12.1 | 8.9 | 16.0 |
| FrozenBiLM (Yang et al., 2022; Xiao et al., 2023a) | ✓ | 1B | 24.2 | 23.7 | 9.6 | 6.1 | 17.5 |
| SeViLA (Yu et al., 2024) | ✓ | 4B | 29.5 | 22.9 | 21.7 | 13.8 | 16.6 |
| *zero-shot (with proprietary LLMs)* | | | | | | | |
| MoReVQA (Min et al., 2024) | ✗ | 340B | 37.8 | 37.6 | 19.7 | 15.4 | 39.6 |
| LLoVi (Zhang et al., 2023a) | ✗ | 1.8T | 37.3 | 36.9 | 20.0 | 15.3 | 24.3 |
| *zero-shot (with open-source LLMs)* | | | | | | | |
| Mistral (Jiang et al., 2023) | ✗ | 7B | 20.4 | 20.2 | 8.7 | 5.9 | 9.2 |
| LLoVi (Zhang et al., 2023a) | ✗ | 7B | 20.7 | 20.5 | 8.7 | 6.0 | 11.2 |
| LLoVi (Zhang et al., 2023a) | ✗ | 12B | 31.4 | 28.8 | 18.4 | 12.0 | 16.2 |
| `LangRepo` (ours) | ✗ | 7B | 20.3 | 20.0 | 8.7 | 6.0 | 11.2 |
| `LangRepo` (ours) | ✗ | 12B | 31.3 | 28.7 | 18.5 | 12.2 | 17.1 |

## 5.2 ABLATION STUDY

**Choice of backbone LLM, text encoder and classifier:** We ablate the choice of LLM-backbones within the framework in Zhang et al. (2023a) in Table 6a. We observe that Mistral-7B (Jiang et al., 2023) is significantly better at video reasoning compared to LLama2-13B (Touvron et al., 2023). Next, we consider different text encoders to embed our text descriptions prior to pruning, such as CLIP-L/14 (Radford et al., 2021) or Sentence-T5-XL (Reimers & Gurevych, 2019) in Table 6b. Surprisingly, CLIP outperforms Sentence-T5 that is trained with a sentence-level objective (which is expected to better align with our caption-similarity computation). Finally, we evaluate different classifiers used for close-ended (*i.e.*, multiple-choice question) VQA setups (see Table 6c). Despite commonly-used in LLM literature, generative classifier performs worse than log-likelihood classifier. Such performance is also intuitive as the latter constrains predictions within the given answer choices (hence, less hallucination). More discussion on this is in supplementary.

**Repository setup and metadata:** In the formulation of `LangRepo` we ablate different hyperparameter settings related to the number of repo-updates (#iterations), the number of video chunks in each iteration (#chunks), and multiple temporal-scales considered when reading data in repository. In Table 6d, we make two observations: (1) more update iterations with finer chunks (higher #chunks per iteration) can preserve more-useful information, and (2) reading information in multiple temporal-scales is consistently better. Moreover, we consider optional metadata to help preserve information that may get lost when pruning (*e.g.* temporal ordering, or repetitive captions), namely, timestamps and #occurrences (*i.e.*, the number of captions grouped within each repo description). We see in Table 6e that #occurrences help weigh each description when summarizing, resulting in better performance. However, timestamps do not provide meaningful improvement in our setup, in the context of EgoSchema VQA.

Table 6: Ablating design decisions on EgoSchema (Mangalam et al., 2024): We evaluate different design decisions of our framework on EgoSchema 500-video subset for zero-shot video VQA.

(a) **Choice of LLM**: In the LLoVi framework, Mistral outperforms LLama2 even at a smaller scale.

| LLM | Scale | Acc. |
|---|---|---|
| Llama2 (Touvron et al.) | 13B | 43.0 |
| Mistral (Jiang et al.) | 7B | 50.8 |

(b) **Text encoder**: CLIP outperforms Sentence-T5 (trained with setntence objective) for similarity-based pruning.

| Text encoder | Acc. |
|---|---|
| Sentence-T5-XL (Reimers & Gurevych) | 56.4 |
| CLIP-L/14 (Radford et al.) | 57.8 |

(c) **VQA classifier**: Log-likelihood classifier performs better on close-ended VQA.

| VQA classifier | Acc. |
|---|---|
| Generative | 57.8 |
| Log-likelihood | 60.8 |

(d) **Repository setup**: Having more iterations with finer chunks in writing, and multiple scales in reading is better in `LangRepo`.

| #Iter | #Ch | Read | Acc. |
|---|---|---|---|
| 1 | [2] | 1 | 57.0 |
| 1 | [4] | 1 | 60.8 |
| 3 | [4,3,2] | 1 | 58.4 |
| 3 | [4,3,2] | 2 | 59.4 |
| 3 | [4,3,2] | 3 | 61.2 |

(e) **Metadata in repository**: Timesteps do not help, yet #occurrences help with proper weighing.

| Model | Acc. |
|---|---|
| `LangRepo` (ours) | 60.8 |
| + tstmp | 60.4 |
| + occ | 61.4 |
| + tstmp + occ | 58.2 |

(f) **Efficiency in a multi-query setup**: Despite being initially expensive, re-using our concise representation on multiple-queries is efficient (measured on an A5000 GPU).

| Model | Params | Latency per video (s) | | |
|---|---|---|---|---|
| | | q/v = 1 | q/v = 2 | q/v = 5 |
| LLoVi (Zhang et al.) | 7B | 22.11 | 44.34 | 108.75 |
| `LangRepo` | 7B | 30.98 | 37.46 | 56.90 |
| LLoVi (Zhang et al.) | 12B | 50.06 | 99.84 | 249.95 |
| `LangRepo` | 12B | 85.09 | 94.90 | 124.33 |

(g) **Captioner**: Clip-level captions (*e.g.* LaViLa) performs better than frame-level ones. A gap to oracle exists.

| Captions | Acc. |
|---|---|
| BLIP-2 (Li et al.) | 55.4 |
| LLaVA-1.5 (Liu et al.) | 58.4 |
| LaViLa (Zhao et al.) | 60.8 |
| Oracle | 69.2 |

(h) **Video input**: Feeding short captions chunk-by-chunk to the LLM is empirically-better than feeding all-at-once.

| Streaming setup | Acc. |
|---|---|
| LLoVi (Zhang et al.) | 50.8 |
| Chunk-based LLoVi | 57.8 |
| `LangRepo` (ours) | 60.8 |

(i) **Input length**: Both Mistral and LLoVi drops performance with increasing input length, whereas `LangRepo` stays more-stable.

| Model | 0.5× | 1× | 2× |
|---|---|---|---|
| Mistral (Jiang et al.) | 49.8 | 48.8 | 46.8 |
| LLoVi (Zhang et al.) | 57.2 | 55.4 | 53.6 |
| `LangRepo` | 56.4 | 57.8 | 56.4 |

**Efficiency in a multi-query setup:** We also ablate the efficiency of our concise representation in Table 6f. `LangRepo` can be initially expensive, as it requires multiple write-read operations (yet, each processing smaller context-lengths). However, once repository is created, it can be re-used more-efficiently in a setup with multiple-queries for a given video (*i.e.*, the initial cost will be amortized). This is especially relevant in practical scenarios, where users may have multiple queries correponding to a given video.

**Captioner quality:** In Table 6g, we evaluate the quality of captions consumed by `LangRepo`. By default, we use short-clip captions from LaViLa (Zhao et al., 2023c), which outperform frame-level captions (BLIP-2 (Li et al., 2023b), LLaVA-1.5 (Liu et al., 2023)). Oracle captions from Ego4D show the performance upper-bound.

**Input format and length:** We consider different ways of consuming long video data, either as a whole or as chunks (see Table 6h). Among these options, processing as chunks enables preserving more fine-grained details in LLM outputs. Our repository setup provides further improvement showing its effectiveness over the baseline with the same chunk-based processing. Finally, we re-visit the experiment on how the input length affects the effectiveness of LLMs, presented in Table 1. In Table 6i, we show that `LangRepo` provide more-stable performance with increasing input lengths, in contrast to baselines.

# 6 CONCLUSION

In this paper, we introduced a Language Repository (`LangRepo`), which reads and writes textual information corresponding to video chunks, as a concise, multi-scale and interpretable language representation, together with additional metadata. Both our `write-to-repo` and `read-from-repo` operations are text-based and implemented as calls to a backbone LLM. Our empirical results show the superior performance of `LangRepo` on multiple long-video reasoning benchmarks at its respective scale, while also being (1) less-prone to performance drops due to increasing input lengths, and (2) interpretable, enabling easier human intervention if and when needed.

## REPRODUCIBILITY STATEMENT

We use open-source LLMs (w/ publicly-available code and pretrained-weights) in all our experiments. By relying on LLMs with reasonable-scale (*i.e.*, not proprietary, paid LLMs), we make our work more-accessible. As all our experiments are done in zero-shot settings, we do not update any pretrained weights. All our evaluations are conducted on publicly-available standard long-video benchmarks. We detail all required steps, and provide prompts to reproduce the proposed contributions. Finally, we pledge to release our code together with the paper to support further research.

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

# A   APPENDIX

## A.1   DESIGN DECISIONS

**Similarity-based pruning:** We notice that the short captions generated by the VLLM can be highly-redundant, as it has a limited temporal span. Such excess details can adversely affect the performance (see Table 1), while also wasting the LLM context. This motivates us to prune redundancies. We consider prompting the LLM directly to identify and rephrase redundant information. However, the outputs in this setup can be noisy and lack of any structure that is useful for parsing. In other words, although redundancies get pruned, there is limited controllability and inability of identifying what gets pruned. Hence, we decide to delegate the function of identifying redundancies to a separate module: a similarity-based grouping with the help of text embeddings. This gives more control on what to prune and how much to prune, while generating outputs that can be parsed to extract other useful metadata (*e.g.* timestamps).

**Processing videos as chunks:** Our decision to consume longer videos as chunks is motivated by prior work (Wu et al., 2022; Ryoo et al., 2023). It allows us to not lose short-term details, while also keeping track of long-term dependencies via multi-scale processing. Additionally, although not explored in the scope of this paper, such a setup integrates well with temporally-fine-grained prediction tasks, where an LLM needs to make multiple predictions over time.

**Choice of metadata:** To avoid the loss of important details during pruning, we maintain additional metadata in our `LangRepo`. Since captions across time can be grouped together in a single repo description, we save their timestamps as a separate field. This can help with temporal reasoning questions. We also update an occurrence counter, which shows the number of captions grouped within a single description. This can act as a weight, to help in cases such as counting or identifying repetitive events.

**All-textual repository:** Instead of being a latent representation (Wu et al., 2022; Ryoo et al., 2023; Balažević et al., 2024), our `LangRepo` is all-textual. This promotes interpretability for human observers, while also being a more-natural form of structure for LLM-based processing. Additionally, our implementation can be formulated to be zero-shot, without requiring any training or finetuning.

**Classifier for close-ended VQA:** The standard multiple-choice question-answering setup considers a generative classifier. Meaning, an LLM is prompted to generate the correct answer option among multiple-choices, directly as next-token prediction. Another approach used in NLP literature is log-likelihood based classification (see Cloze prompting in (Robinson et al., 2023)). Here, the LLM is prompted separately for each of the multiple choices with a template such as "`Question: Answer-option`". The choice that maximises the log-likelihood of predicted tokens (*i.e.*, tokens corresponding to `Answer-option`) is selected as the correct answer. This is a more-natural setup for close-ended VQA since it avoids hallucination. Among these classifiers, we find the latter to be better-performing. Yet, it is more-sensitive to the prompt template. We direct the reader to supplementary A.2 for more details.

## A.2   PROMPTING FOR VQA

As the evaluation setup, we consider multiple-choice visual question-answering (VQA) on long videos. Given the close-ended answer formulation, we can consider two different classifiers to make the prediction: (1) a Generative classifier, which directly generates the answer choice, or (2) a Log-likelihood classifier, which select the most-probable choice based on the joint-probability of tokens in each answer option given the description and the question. As we discussed in Sec. A.1, the latter generally performs better, as it is less-prone to hallucinations (*i.e.*, prediction is explicitly constrained to answer choices). However, it is also sensitive to the prompts we use. Hence, we include a discussion on prompting in the following subsections.

**Generative classifier:** Here, we direcly prompt the LLM to generate the correct answer, conditioned on the descriptions generated by `LangRepo`, the question and the answer options (inspired by (Zhang et al., 2023a)). To make sure that the output can be parsed, we provide additional guiding instructions and any syntax specific to the LLM (Mistral (Jiang et al., 2023)). This also discourages any hallucinations. On all benchmarks, we use the common prompt given below.

```
``[INST] <<SYS>> You are a helpful expert in first person view video anal-
ysis. <</SYS>> Please provide a single-letter answer (A, B, C, D, E) to
the following multiple-choice question, and your answer must be one of
the letters (A, B, C, D, or E). You must not provide any other response or
explanation. You are given some language descriptions of a first person
view video.  The video is ${duration} seconds long.  Here are the de-
scriptions: ${description}.\n You are going to answer a multiple choice
question based on the descriptions, and your answer should be a single
letter chosen from the choices.\n Here is the question: ${question}.\n
Here are the choices.\n A: ${optionA}\n B: ${optionB}\n C: ${optionC}\n
D: ${optionD}\n E: ${optionE}\n [/INST]''
```

**Log-likelihood classifier:** In this setup, we prompt the LLM with each answer option separately, and select the highest-probable answer. The probability is computed only on the tokens of the answer option, conditioned on the input sequence. In our experiments, we notice that the effectiveness of this method is sensitive to the prompt. This is due to the question-answer formats in the dataset considered. For instance, EgoSchema (Mangalam et al., 2024) consists of full-sentence answers, whereas NExT-QA (Xiao et al., 2021) consists of answer phrases. Hence, the latter benefits from additional guidance from formatting within the prompt template. More specifically, on EgoSchema (Mangalam et al., 2024), our prompt has the following format.

```
``${description} ${question} ${answer_option}''
```

Here, the probability is computed only on ${answer_option}. However, on the benchmarks based on NExT-QA (Xiao et al., 2021) data, our prompt has the following format with more structure.

```
``${description} Based on the description above, answer the follow-
ing question: ${question}?  Select one of these choices as the an-
swer:\n A: ${optionA}\n B: ${optionB}\n C: ${optionC}\n D: ${optionD}\n
E: ${optionE}\n The correct answer is, ${option_id}: ${answer_option}''
```

Here, the probability is computed only on ${option_id}:  ${answer_option}. We observe that neither prompt template works as effective when interchanged.

## A.3  QUALITATIVE EXAMPLES OF REPOSITORY ENTRIES

We present qualitative examples from EgoSchema (Mangalam et al., 2024) dataset to better clarify the operations in LangRepo. In Fig. 4, we show the format of repository entries. Here, non-redundant captions from the input get directly written to the repo. In contrast, any redundant captions— grouped based on similarity— get rephrased as concise descriptions (1 per-group). Each repository description may come with additional metadata such as timestamps and #occurrences to avoid the loss of meaningful information due to pruning. In Fig. A.1, we further elaborate on multiple scales within the repository, which are generated by iteratively processing increasingly-longer chunks (created by re-chunk operation). During reading, we can decide to summarize information at various temporal scales to generate output descriptions useful for VQA.

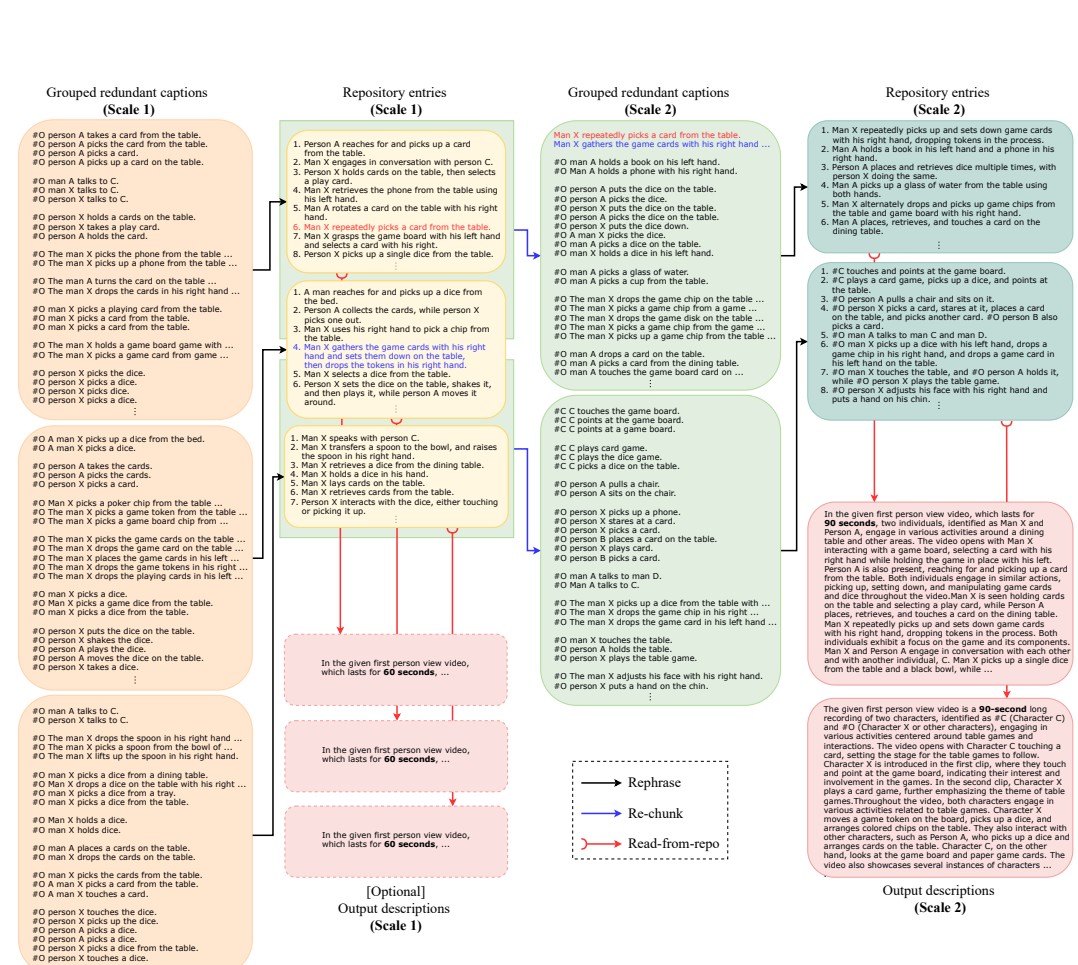

Figure A.1: **A qualitative example of iterative writing and multi-scale reading in `LangRepo`:** Here, we present an example with 2-scales, given captions of a 180s long video. In scale-1, we consider 3 chunks of 60s each, and in scale-2, we re-chunk them into 2 chunks of 90s each. We only show the redundant captions that go through pruning, and also, omit any metadata (*e.g.* timestamps) within the repository. In each scale, captions grouped based on similarity get rephrased concisely. To generate inputs of the subsequent scale, we simply order previous repository descriptions in time, and split (i.e., re-chunk) into fewer (and, longer) chunks. When reading, each entry in each scale is summarized separately to create output descriptions of various temporal spans. In general, we always consider the last-scale descriptions to be mandatory, but any prior-scale to be optional. Yet, we observe multiple scales to be beneficial (see Table 6d). Best-viewed with zoom-in.

