# OpenReview forum: "Language Repository for Long Video Understanding"
_ICLR.cc/2025/Conference — Submitted to ICLR 2025_

### Official Review · Reviewer_Mbnb · 2024-10-31

**Soundness:** 2
**Presentation:** 2
**Contribution:** 1
**Rating:** 3
**Confidence:** 5

**Summary:**

The paper presents a Language Repository for LLMs, which generates multi-scale summaries for the video input and applies in long video understanding. The authors evaluate the proposed LangRepo on EgoSchema, NExT-QA, IntentQA and NExT-GQA benchmarks.

**Strengths:**

1. The authors include enough relevant papers in the related works section.
2. The authors provide a detailed ablation study.

**Weaknesses:**

1. The contribution of novelty is concerning. In this paper, the authors propose a summarization-based method based on the dense caption from LLoVi [1] paper. However, the original LLoVi paper already proposes a summarization prompt which significantly improves the performance of long video QA tasks. But in this paper, the authors do not have any discussion on this point in related work or any other sections. Also, the “Grouping redundant text” section is similar to the “Caption Digest” in the LifelongMemory[2] paper. The authors should provide strong evidence of the paper’s novel contribution compared to the existing works.

2. The need for complicated textual operation is questionable. In this paper, the authors only showcase the results on open-source LLM on a 7B/13B scale and the EgoSchema fullset results are significantly lower than the proprietary LLMs models (even comparing to the same scale open-source VLMs, like InternVideo2, is lower). The potential need of the summarization module is only because the LLM backbone is weak and could not take in the dense caption list. Thus, the authors should validate the effectiveness of the proposed framework on strong proprietary LLMs and have fair comparison with those methods.

3. The use of “Log-likelihood classifier” is confusing and creates unfair comparison. The authors mention that the Log-likelihood classifier is only used for EgoSchema dataset and improves significantly to the subset performance. In that case, since most methods in the main tables are using the lower-performance “generative classifier”, the use of this classifier creates unfair comparisons for the most related works. Also, the results on the EgoSchema subset are significantly lower than the subset (over 20%), this is not intuitive since the subset is just a part of the full set. Could the authors elaborate more on this?

4. The evaluation of “long video” is not sufficient. The authors provide results on NExT-QA, IntentQA and NExT-GQA benchmarks, however, the average video length of these dataset is only about 44 seconds, which is considerably short. The authors should provide results on more than 10 minutes to hour long video benchmarks to showcase the effectiveness of the proposed method on long videos.

5. The efficiency of the proposed method is very concerning. Since the multi-round summarization process is quite slow, and in Table 6.f, compared to the baseline approach LLoVi, LangRepo consumes 70% more time per video-query pair, using 85 seconds to process a 180 seconds video from egoschema. The authors should also show the detailed split of times using on captioning/summarization/question-answering etc..

[1] A Simple LLM Framework for Long-Range Video Question-Answering

[2] LifelongMemory: Leveraging LLMs for Answering Queries in Long-form Egocentric Videos

**Questions:**

All questions from the weaknesses, and:

1. The comparison in the right part of Figure 1 is concerning, since the methods like VideoAgent only uses only 8.4 frames, the authors should take the caption/frame numbers into account (also main tables).
2. Could the authors test the model with the more advanced and larger LLMs, like qwen-72B or LLaMA-3.1-70B?
3. Could the authors provide comparison results in main table using "generative classifier" for the proposed method as well for fair comparison?

---

> ### Author Response · Authors · 2024-11-23
> **Response to reviewer Mbnb [1/4]**
>
> **W1.1: Clarification on novelty (compared to summarization in LLoVi).**
>
> We thank the reviewer for raising this concern. LangRepo introduces multiple novel contributions compared to LLoVi:
>
> (1) LangRepo (in its write operation) **explicitly removes redundant content given in text**, which is not explored in LLoVi. Due to this reason, LangRepo shows a better LLM context utilization with increasing input lengths, resulting in stable and better performance (see table below– also in Table 6i). In contrast, LLoVi show a degraded performance at longer inputs--- a behavior that is also observed in literature [1]. The benefit of the redundancy-removal can also be seen in Table 6h, as the comparison between chunk-based LLoVi (57.8%) vs. LangRepo (60.8%).
>
> | Model 		| 0.5x 	| 1x 	| 2x |
> |----------|----------|----------|----------|
> | LLoVi 		| 57.2 	| 55.4 	| 53.6	|
> | LangRepo 	| 56.4 	| 57.8 	| 56.4 	|
>
> (2) LangRepo (with its iterative application of write and read) **creates multi-scale temporal information expressed as text**. Our read operation (similar to LLoVi), focuses on each temporal scale separately, extracting varying levels of temporal dependencies (as shown in the qualitative example in Fig A.1— dependencies across 60s vs. 90s windows). The impact of such multi-scale information in LangRepo can be seen in Table 6d, where multiple scales (*i.e.,* #iter) improves the performance (60.8% -> 61.2%).
>
> (3) LangRepo **creates an intermediate representation (*i.e.,* repository entries) that is fully text-based (*i.e.,* human-readable), concise and structured**. Such a representation is not available in LLoVi. This is an effective way of storing long-video information, which is efficient to be used with repeated queries — as shown in Table 6f (2x speedup with 5-queries per video). Moreover, we preserve additional metadata (*e.g.* timestamps, #occurrences) within our representation, that can be useful in the downstream. For instance, #occurrences help improve the performance (60.8% -> 61.4%) as shown in Table 6e.
>
> Based on the above, we believe LangRepo provides a meaningful advancement over a fully text-based video framework such as LLoVi, that simply does summarization of video captions. We will better clarify our novelties in the final version of the paper.
>
> [1] Same Task, More Tokens: the Impact of Input Length on the Reasoning Performance of Large Language Models [arXiv 2024]
>
>
> **W1.2: Clarification on novelty (compared to Caption Digest in LifelongMemory).**
>
> We thank the reviewer for raising this concern. LifelongMemory includes a *’Caption Digest’* module which shows conceptual similarities (*e.g.* removing redundancies) to grouping/rephrasing in LangRepo. However, there are notable and practically-meaningful differences in the execution.
>
> Lifelong Memory digest captions by first picking captions relevant to a given query (*i.e.,* query-conditioning), and then merging **only adjacent** captions that are similar. In contrast, LangRepo’s grouping/rephrasing (1) **is not query-conditioned**, and (2) **removes redundancies across any point in time** (*i.e.,* potentially reducing redundancies even further).
>
> The above (1) has practical impact. Since the creation of our intermediate representations (*i.e.,* repository entries) does not rely on a specific query, it can be re-used across multiple queries on the same video more efficiently (*i.e.,* rephrasing are better generalizable).
>
> The above (2) is also important as LangRepo will be more-effective in context-utilization, giving better performance at a comparable setting. To further validate this, we compare LifelongMemory-style caption digest with our framework, based on the results report in their original paper (see table below). LangRepo outperforms LifelongMemory at a similar scale (60.8% vs. 60.4%) and scales much better (66.2% at 12B vs. 64.0% at 175B).
>
> | Model 			| Redundancy removal 	| Params 	| ES-subset Acc % |
> |----------|----------|----------|----------|
> | LifelongMemory 	| Caption Digest		| 8B 		| 60.4	|
> | LifelongMemory 	| Caption Digest 	| 175B 		| 64.0 	|
> | LangRepo 		| Group/Rephrase 	| 7B 		| 60.8	|
> | LangRepo 		| Group/Rephrase 	| 12B 		| 66.2 	|
>
> Furthermore, we also highlight that LifelongMemory provides a very limited discussion on their caption digest, and no qualitative visualizations on how-well it works. In contrast, LangRepo provides an extended discussion (Fig 2-left, Fig 3-top and L242-295, L469-472, L922-931), evaluation (Table 6b, 6h and 6i) and qualitative samples (Fig 4 and A.1), giving reader a holistic idea about this concept and its ramifications in effective context-utilization.

---

> > ### Author Response · Authors · 2024-11-23
> > **Response to reviewer Mbnb [4/4]**
> >
> > **W4: LangRepo performance on very-long video (*e.g.* Short Film Dataset) question-answering.**
> >
> > We thank the reviewer for bringing this benchmark comparisons to our attention. We agree that evaluations on very long-video QA show the generalization of LangRepo. Hence, in this rebuttal, we include results on ShortFilms Dataset (average length of ~13mins).
> >
> > | Model | Temporal window | SFD Acc % (change) |
> > |----------|----------|----------|
> > | FrozenBiLM	| Scene-level	| 22.7		|
> > | FrozenBiLM	| Movie-level	| 23.4 (+0.7)	|
> > | LLoVi		| Scene-level	| 34.2		|
> > | LLoVi		| Movie-level	| 30.8 (-3.4)	|
> > | LangRepo	| Scene-level	| 26.9		|
> > | LangRepo	| Movie-level	| 29.0 (+2.1)	|
> >
> > Here, the authors of the dataset paper already included LangRepo as one of their standard baselines for very long-video QA. This fact validates that LangRepo is being already adopted in the community. In SFD, various temporal window-sizes (shot-level -> scene-level -> movie-level) are considered for evaluation. We present an interesting observation (see the table above), when going from scene-level to movie-level, that corresponds to very long-video QA with *visual information given as text*— the closest setup to our original LangRepo evaluation.
> >
> > Even without any hyperparameter tuning, LangRepo performs competitively with similar baselines (FrozenBiLM, LLoVi), while also showing a striking difference when going up to movie-level (+2.1%), compared to LLoVi (-3.4) and FrozenBiLM (+0.7). This shows third-party evidence that LangRepo can generalize to very long-videos. We will include this discussion in the final version of the paper.
> >
> >
> > **W5: On the efficiency of LangRepo.**
> >
> > We agree that the write and read operations (*i.e.,* multi-round redundancy-removal and summarization) of LangRepo has a slower inference speed compared to a simpler summarization technique such as LLoVi.
> >
> > However, we also highlight how the intermediate representation that we create in LangRepo (*i.e.,* repository entries) can be efficient in practice (see Table 6f). Such a representation— that is not available with LLoVi— is an effective way of storing long-video information concisely, which can be reused repeated queries. Although LangRepo is 70% slower compared to LLoVi with 1-query per video, it matches the speed (1x) with 2-queries per video, while being **2x faster with 5-queries per video**.
> >
> > Such settings with repeated queries are practically-relevant. For instance, many long-video QA benchmarks include multiple-queries per video (*e.g.* 8.76 q/v in NExT-QA, 3.76 q/v in IntentQA, 5.61 q/v in NExT-GQA). Here, having a concise representation that can be queried efficiently would be highly useful, rather than having to re-process all the information from scratch. We will expand the discussion in the paper (L513-518) to better-highlight such use-cases.
> >
> >
> > **Q1: Is it better to compare Performance vs. Frame-count in Fig 1 (instead of Performance vs. Model size)?**
> >
> > We understand this point-of-view, but we have to respectfully disagree. The frame count does not necessarily translate to a practical complexity measure (either computational-cost or model-capacity). Let us elaborate below.
> >
> > Having fewer-frames considered for the question-answering, does not necessarily mean that the corresponding pipeline is computationally-efficient or faster in terms of inference speed. For instance, even though VideoAgent [arXiv 2024] rely on 8.4 frames on average, their discussion on inference speed (in Appendix A of the corresponding paper) shows that it takes ~35s to run iterative inference on a single video (or, ~200s if iterative captioning of newer 8.4 frames is included). In comparison, processing 180 frames with LangRepo takes ~85s. Although this is not a direct comparison, we want to highlight that having 8.4 frames (instead of 180) does not necessarily translate to considerable inference speedups (or computational savings). Another example is the concurrent work LVNet [arXiv 2024], which relies on just 12 frames for question-answering on EgoSchema. However, this method initially processes 900 frames, in order to select the final 12 frames, which is not necessarily efficient.
> >
> > On the other hand, recent literature on LLMs heavily rely on the parameter count as the primary complexity measure of the model capacity (*e.g.* how-well they perform on benchmarks, including question-answering). We believe this provides a better metric to compare different models with, and extrapolate how generic pipelines (*e.g.* LangrRepo, LLoVi) can scale with larger models. Due to these reasons, we believe a comparison w.r.t. the model size is more-meaningful.

---

> ### Author Response · Authors · 2024-11-23
> **Response to reviewer Mbnb [2/4]**
>
> **W2.1: Performance of LangRepo on EgoSchema-fullset compared similar-sized models (7B).**
>
> We understand the concern of the reviewer on this point. However, we wish to highlight that, a direct comparison with single-stage VLMs that are also pretrained with video-captions— even at a comparable scale— is not reasonable, as such models have seen large-scale video/temporal information. In comparison, zero-shot two-stage frameworks (*e.g.* LLoVi, LangRepo) has pure-LLMs performing video QA, while having *no video-caption pretraining*. We highlight the disparity of these two setups with *‘video pretrain’* column in all our main result tables (also included below). In comparable settings, LangRepo shows a competitive performance on EgoSchema fullset.
>
> | Model 		| Video-pretrain 	| Params	| ES-fullset Acc % |
> |----------|----------|----------|----------|
> | Tarsier 	| Yes		| 7B 		| 49.9	|
> | VideoChat2 	| Yes 		| 7B 		| 54.4 	|
> | mPLUG-Owl 	| No 		| 7B 		| 31.1	|
> | MVU 		| No 		| 7B 		| 33.5 	|
> | LLoVi 		| No 		| 7B 		| 37.6 	|
> | LangRepo 	| No 		| 7B 		| 38.9 	|
>
>
>
> **W2.2: Performance of LangRepo on EgoSchema when scaling to larger LLMs.**
>
>
> First, we want to highlight that in this paper, we experiment with open-source LLMs to keep the full LangRepo pipeline accessible, and to avoid significant monetary costs of using proprietary models (*e.g.* estimated ~$200 per EgoSchema fullset experiment with GPT-4o). Fully-text based methods such as LLoVi has already shown the scaling properties of this class of models (*e.g.* with GPT-3.5), and we expect a similar scaling behavior with LangRepo.
>
> To still support this claim with open-source models, in this rebuttal, we run a new experiment with the larger LLMs  (*e.g.* Llama3.1-70B), showing a considerable performance boost on EgoSchema. We will include this discussion in the paper.
>
> | Model 			| Params	| ES-subset Acc % |
> |----------|----------|----------|
> | LifelongMemory 	| 8B 		| 60.4		|
> | LifelongMemory 	| 175B 		| 64.0 		|
> | LLoVi 			| 7B 		| 50.8 		|
> | **LLoVi** 		| **70B** 	| **62.2** 	|
> | LangRepo 		| 7B 		| 60.8 		|
> | **LangRepo** 	| **70B** 	| **67.0** 	|
>
> *all new numbers are in bold.*
>
> **W2.3: Is the need for a summarization technique only due to a weaker LLM?**
>
> No, such summarization techniques (*e.g.* as in LLoVi, LangRepo) are even useful in larger LLMs with longer context lengths.
>
> (1) First, frameworks such as LLoVi has already shown the benefits of summarization even with very large proprietary models (eg: GPT-3.5). We expect a similar scaling behavior with LangRepo (as also validated by Llama3.1-70B experiment above).
>
> (2) Moreover, the underlying problem that we address in this paper— decaying effectiveness of LLMs with increased input lengths— is also observed with much larger models (*e.g.* Gemini) in the recent concurrent literature [1].
>
> (3) Finally, we want to highlight that the Mistral/Mixtral backbones that we primarily rely-on are not weak, and definitely handle the full dense captions without any context overflow/truncation. For instance, a typical EgoSchema dense captions contain ~1.8k tokens— that is reduced to ~0.7k tokens by LangRepo— which can be handled comfortably within the context length of these models (8k in Mistral, 32k in Mixtral). Therefore, we believe LangRepo framework can be useful irrespective of the model scale and the context length. We will better highlight this in the final version of the paper.
>
> [1] Same Task, More Tokens: the Impact of Input Length on the Reasoning Performance of Large Language Models [arXiv 2024]

---

> ### Author Response · Authors · 2024-11-23
> **Response to reviewer Mbnb [3/4]**
>
> **W3.1: Do main results tables make a fair comparison (w/ Log-likelihood classifier)?**
>
> We sincerely apologize for the confusion. We use the LL-classifier in *all* our main benchmark experiments (*e.g.* EgoSchema, NExT-QA, IntentQA, NExT-GQA), as we consider it to be an inclusive component of the LangRepo framework. Such usage is also discussed in the supplementary  (L983-1003), when we introduce specific prompting styles with the LL-classifier for each underlying dataset.
>
> To make sure we still do a fair comparison, the results corresponding to the closely-related baselines (*e.g.* Mistral, LLoVi) that we already include in our main tables (Table 2-5), use the same LL-classifier setup (instead of Gen-classifier). However, replicating the results for all other baselines under LL-classifier is prohibitively-expensive. For better clarity, we additionally provide the performance on EgoSchema with the two different classifiers in the table below, extending the results in Table 2 and 6c. We also note that other concurrent work (*e.g.* TOPA [arXiv 2024], MVU [arXiv 2024]) also adopt LL-classifier for MCQ benchmarks. We will better clarify the evaluation setting for each experiment in the final version of the paper, avoiding such confusion.
>
>
> | Model 			| w/ LL-classifier		| w/ Gen-classifier |
> |----------|----------|----------|
> | Mistral (7B) 		| 48.8 			| **47.2**	|
> | LLoVi (7B) 		| 50.8 			| **50.2** 	|
> | LangRepo (7B)	| 60.8 			| 58.8 		|
>
> *all new numbers are in bold.*
>
>
>
> **W3.2: Large gap between EgoSchema fullset and subset performance.**
>
> We understand this valid concern. Such a performance gap between the fullset and the subset of EgoSchema is also observed in many prior/concurrent work. Moreover, it is not only prevalent in smaller models (*e.g.* 7B, 12B), but also can be seen in larger models as well (*e.g.* 175B, 1.8T)— even though the gap is smaller (see the table below). We believe the limited generalization of smaller models result in the wider gap.
>
> | Model 			| Params	| ES-Subset Acc %	| ES-fullset Acc % 	| Gap 	 |
> |----------|----------|----------|----------|----------|
> | MC-ViT-L 		| 424 M		| 62.6		| 44.4		| **-18.2** 	 |
> | LongViViT 		| 1B		| 56.8		| 33.3		| **-23.5**	 |
> | LLoVi			| 175B		| 57.6		| 50.3		| **-7.3** 	 |
> | VideoAgent 		| 1.8T		| 60.2		| 54.1		| **-6.1**	 |
> | VideoChat2 		| 7B		| 63.6		| 54.4		| **-9.2**	 |
> | MVU	 		| 7B		| 60.3		| 37.6		| **-22.7**	 |
> | TOPA			| 7B		| 64.5		| 41.7		| **-22.8**	 |
> | LLoVi			| 7B		| 50.8		| 33.5 		| **-17.3**	 |
> | LangRepo		| 7B 		| 60.8		| 38.9 		| **-21.9**	 |
>
> Moreover, in recent work TOPA [arXiv, May 2024], this performance gap is attributed to considerable linguistic changes between QA pairs in the subset and the fullset. A qualitative sample from the same paper is given below for clarification, which highlights the different structures in verbs and phrases in answer choices. We will include this discussion in the supplementary.
>
> [EgoSchema subset]
>
> *Question: Can you summarize the primary objective and the steps the person took throughout the video to achieve it? ensure your answer captures the essence of the video without listing all actions.*
>
> *Answer option 1: The main aim of the person’s primary objective was to **create and build** a new, sturdy wooden bench.*
>
> *Answer option 1: The primary objective for the person was to thoroughly **repair and restore** the wooden bench.*
>
> *Answer option 1: The person’s primary objective was to thoroughly **clean and sanitize** the wooden bench’s surface.*
>
> [EgoSchema fullset]
>
> *Question: Considering the entire video, what would you identify as the most crucial moments in the person’s shopping experience and why?*
>
> *Answer option 1: **Following** astrict shopping list as a guideline and **rejecting** unfit produce.*
>
> *Answer option 1: **Conducting** taste tests and **checking** for the freshness of each vegetable.*
>
> *Answer option 1: **Using** math algorithm for optimal vegetable selection.*

---

> ### Author Response · Authors · 2024-11-25
> **Follow-up**
>
> Dear Reviewer Mbnb,
>
> Thank you again for your constructive feedback and time/effort reviewing our paper. Since the rebuttal period is ending soon, please let us know if our responses have addressed your concerns. We are happy to engage in further discussion to provide more clarifications if needed.
>
> Kind Regards!

---

> > ### Author Response · Authors · 2024-11-27
> > **Follow-up (2)**
> >
> > We thank the reviewer Mbnb again for the initial feedback, which will definitely improve the quality and clarity of this paper. We believe we have addressed all of the reviewer's concerns, but we would be very happy to engage in further discussion and provide more clarifications if needed. Please let us know if the concerns have been addressed.
> >
> > Thanks so much!

---

> ### Comment · Reviewer_Mbnb · 2024-11-27
>
> I would like to thanks to the authors for the response. I think several of my concerns are still not solved.
>
> 1. For W2.3 I think I am still not convinced, is it possible for the authors to conduct experiment on GPT-4 to verify the summarization method proposed by LangRepo still works?
>
> 2. For W4, could the authors test the proposed method on a comprehensive evaluation benchmark with even longer videos  (like Video-MME/LongVideoBench) instead a domain-specific dataset?
>
> 3. For W5, even for the multi-query scenarios, is it possible for the authors to prove that LangRepo already extracted exclusive information for all kinds of information from the videos and do not need to look into the video again? Since the captioner for EgoSchema is LaViLA, which is rather coarse-grained and give a question about the fine-grained detail, it is highly possible that the captioner miss the detail and LangRepo as well.
>
> 4. For Q1, I think if the authors believe the caption number should compare with inference time, I highly encourage the authors to incorporate a table comparing the cost of time/caption number and all related features to give a more clear view of the efficiency and effectiveness of the proposed method comparing to more existing methods.

---

> ### Author Response · Authors · 2024-12-02
> **Follow-up response to reviewer Mbnb**
>
> We really appreciate the engaged discussions from the reviewer, and we are happy to provide more clarifications.
>
> **F-Q1: Is the summarization still effective with larger models (*e.g.* GPT-4)?**
>
> As per the reviewer’s suggestion, we conduct a GPT-4 scale experiment with LangRepo and report the observations in the table below. Here, we follow the same experimental setup as in LLoVi (*i.e.,* using generative classifier), to make a fair comparison with the GPT-4 based results reported in the same paper.
>
> We see a clear benefit of using summarization techniques for long-video QA, over a pure GPT-4 baseline: both in LLoVi (+2.2%) and in LangRepo (+5.6%). This further validates that LangRepo (and its summarization) is still effective with larger LLMs.
>
> | Model 		| Params.	|EgoSchema Acc % 	|
> |----------|----------|----------|
> | GPT-4		| 1.8T		| 59.0			|
> | LLoVi		| 1.8T		| 61.2 (+2.2)		|
> | LangRepo	| 1.8T		| **64.6 (+5.6)**		|
>
>
> **F-Q2: LangRepo performance on more-generic very long-video benchmarks (*e.g.* LongVideoBench).**
>
> As per the reviewer’s suggestion, we conduct a LangRepo experiment on a more-generic, very long-video benchmark: LongVideoBench validation split (w/ up to 1hr videos). Here, we run a LangRepo framework on captions extracted with LLaVA, evaluating its performance against similar-scaled models. In the table below, we observe that LangRepo shows a competitive performance, validating its effectiveness in very long-video QA.
>
> | Model 				|LongVideoBench Acc %	|
> |----------|----------|
> | LLaVA-1.5			| 40.3				|
> | PLLaVA			| 40.2				|
> | ShareGPT4Video 		| 39.7				|
> | VideoLLaVA			| 39.1				|
> | mPLUG-Owl2			| 39.1				|
> | LangRepo			| **38.2**			|
> | Mistral			| 37.4			|
> | VideoChat2			| 36.0				|
>
>
> **F-Q3: Does the Captioner/LangRepo extract sufficient information to answer various questions in multi-query settings?**
>
> We understand the reviewer’s concern. Let us clarify further.
>
> The captioner (*e.g.* LaViLa or LLaVA) usually extracts generic information that can be relevant to various questions, as it is based on a generic prompt (*e.g.* *“Describe this image.”*). When run at a reasonable frequency (*e.g.* 0.5fps or 1fps), such captioners can extract fine-grained information sufficient for QA, as validated by the strong performance of such caption-based models at their scale (as reported in our main results tables— *e.g.* LLoVi, LangRepo).
>
>
> The descriptions generated by LangRepo can also be generic and sufficient for QA in multi-query settings (as highlighted in tables below). Here, NExT-QA consists of 8.76 queries per video on average, where they can either be *Causal*, *Temporal* or *Descriptive* queries. In IntentQA, we have 3.76 queries per video on average, where they can either be *why?*, *how?* or *before/after* queries. LangRepo shows a strong performance in each different split (*i.e.,* on different questions based on the same video), validating its effectiveness in extracting generic information for videos with multiple-queries.
>
> | Methods on NExT-QA	| Causal (%)  	| Temporal (%)	| Descriptive (%)	|
> |----------|----------|----------|----------|
> | Mistral 		| 51.0		| 48.1		| 57.4			|
> | LLoVi 			| 60.2		| 51.2		| 66.0			|
> | LangRepo 		| 64.4		| 51.4		| 69.1			|
>
>
> | Methods on IntentQA	| why? (%)  	| how? (%)	| before/after (%)	|
> |----------|----------|----------|----------|
> | Mistral 		| 52.7		| 55.4		| 41.5			|
> | LLoVi 			| 59.7		| 62.7		| 45.1			|
> | LangRepo 		| 62.8		| 62.4		| 47.8			|
>
> That being said, we agree with the reviewer that there can be instances where either the captioner or LangRepo fails to capture required information for QA. Yet, the performance above suggests that this happens less-often compared to similar baselines. We will discuss this under the limitations section in the final version of the paper.
>
>
> **F-Q4: Reporting frame-count with LangRepo performance (in addition to model size).**
>
> We agree with the reviewer’s point. For a more-comprehensive view on long-video QA methods, it is important to compare them across different axes.
>
> Hence, based on the  reviewer’s suggestion, we will report a “performance vs. frame-count” comparison (as shown in this new [anonymous-figure](https://drive.google.com/file/d/1iRbj-AHGnZXRQPiDTq4PrP5JiNNkvMdl/view?usp=share_link)), in addition to our current “performance vs. model-size” comparison given in Fig 1-right. We will also include frame-counts for all models in our tables.
>
> Moreover, when discussing the efficiency/effectiveness claims of LangRepo, we will clarify that we state such claims w.r.t. the model-size (rather than the frame-count). We understand that this is an important distinction for the reader, and thank the reviewer for raising this point.
>
> **We hope that we were able to address the reviewer's concerns. We also hope that our additional experiments and clarifications will be kindly considered in the final rating.**

---

> > ### Comment · Reviewer_Mbnb · 2024-12-02
> >
> > Thanks the authors for the detailed response, I think some details/clarification of the results are still missing.
> >
> > For F-Q2, what is the exact model of the captioner LLaVA model (which version)? Also the LongVideoBench contains four split of different length of video, if the results provided by the authors are the overall accuracy, could the authors show that ablation as well? Also for the current results, it seems LangRepo downperforms the LLaVA-1.5 model which is much simple and efficient model, could the authors provide any clarification on this result?
> >
> > For F-Q3, the authors mentioned "The captioner (e.g. LaViLa or LLaVA) usually extracts generic information that can be relevant to various questions ... When run at a reasonable frequency (e.g. 0.5fps or 1fps), such captioners can extract fine-grained information sufficient for QA", however, this argument is confusing since if the captioner are coarse-grain, the dense output captions will just be some repeated general information instead of adding additional object-level or fine-grain information.

---

> ### Author Response · Authors · 2024-12-02
> **Follow-up response to reviewer Mbnb**
>
> > For F-Q2, what is the exact model of the captioner LLaVA model (which version)?
>
> We use LLaVA-1.5 (7B) captioner with single-image input.
>
> > Also the LongVideoBench contains four split of different length of video, if the results provided by the authors are the overall accuracy, could the authors show that ablation as well?
>
> We present the results on each duration-split of the validation set in the table below. LangRepo outperforms Mistral based on the same captioner. LongVideoBench does not report the above baseline results on different validation splits (but only for the test splits). We will include the LangRepo results on the test-splits with the final version of the paper, given the insufficient time remaining in the rebuttal period to conduct new experiments on the test-splits.
>
> | Method $\downarrow$ \ Durations (s) $\rightarrow$		| (8,15]  	| (15,60]	| (180,600]  	| (900,3600]	| All	|
> |----------|----------|----------|----------|----------|----------|
> | Mistral 		| 43.6		| 48.3		| 37.8		| 31.7		| 37.4	|
> | LangRepo 		| 43.1		| 50.6		| 38.8		| 32.3		| 38.2	|
>
>
> > Also for the current results, it seems LangRepo downperforms the LLaVA-1.5 model which is much simple and efficient model, could the authors provide any clarification on this result?
>
> We understand the reviewer’s concern. Due to the difference in the experimental settings (*e.g.*, single-stage vs. two-stage, single-image vs. multi-image), these results are not directly-comparable. Let us clarify below.
>
> The LLaVA-1.5 setup reported in LongVideoBench use this multi-modal LLM with **multi-image input** directly for a **question-answering** task. However, since LangRepo is a two-stage pipeline (*i.e.,* captioner + question-answering LLM), we have to use LLaVA-1.5 with **singe-image input** for a **captioning** task, followed by question-answering. In other words, we rely on its captioning capability, which turned out to be not as good as the QA capabilities.
>
> LongVideoBench includes only single-stage VQA baselines in their paper, which is different from the LangRepo setup. We provide a more-direct comparison in the above table with Mistral (using the same captioner), validating the improvements based on LangRepo.
>
>
> > For F-Q3, the authors mentioned "The captioner (e.g. LaViLa or LLaVA) usually extracts generic information that can be relevant to various questions ... When run at a reasonable frequency (e.g. 0.5fps or 1fps), such captioners can extract fine-grained information sufficient for QA", however, this argument is confusing since if the captioner are coarse-grain, the dense output captions will just be some repeated general information instead of adding additional object-level or fine-grain information.
>
> We agree with the reviewer’s concern, Yet, we highlight that this is a limitation of the captioner, not the techniques that we introduce in LangRepo (redundancy-removal + multi-scale summarization). Such limitations also apply to any two-stage VQA pipeline that relies on a captioner (*e.g.* LLoVi, Mistral). Among these pipelines, we show that LangRepo performs the best.
>
> An advantage of such a two-stage pipeline is that, we can easily plug-in better models for each stage when available (either the captioner, or question-answering LLM), which is not possible with single-stage pipelines (due to required finetuning— at least the projector layers). We will discuss such pros and cons in a separate section.
>
> We thank the reviewer for the time and effort spent on these discussions. We hope that we were able to address the reviewer's concerns. We also hope that our additional experiments and clarifications will be kindly considered in the final rating.

---

> > ### Author Response · Authors · 2024-12-03
> > **Follow-up**
> >
> > Dear Reviewer Mbnb,
> >
> > We thank you again for your initial feedback and continued engagement in discussions. Please let us know if your concerns have been addressed. Since the rebuttal period is ending soon, we would really appreciate it if our additional experiments and clarifications can be considered in the final rating.
> >
> > Thanks so much!

---

> > > ### Author Response · Authors · 2024-12-03
> > > **End of discussion-period follow-up**
> > >
> > > Dear Reviewer Mbnb,
> > >
> > > We thank you again for your initial feedback and continued engagement in discussions. Since the discussion period has ended, we would greatly appreciate it if our additional experiments and clarifications could be considered in your final rating.
> > >
> > > Thanks so much!

---

### Official Review · Reviewer_VDGU · 2024-11-03

**Soundness:** 3
**Presentation:** 3
**Contribution:** 2
**Rating:** 5
**Confidence:** 2

**Summary:**

This paper introduces a Language Repository (LangRepo) for LLMs, addressing the challenge of decreased effectiveness with longer inputs. LangRepo maintains concise and structured information, updating iteratively with multi-scale video chunks. It includes operations to eliminate redundancies and extract information at various temporal scales. The framework demonstrates state-of-the-art performance on zero-shot video question-answering benchmarks.

**Strengths:**

1. The authors propose an iterative approach with progressively longer chunks, enabling LangRepo to learn high-level semantics and extract stored language information at various temporal scales. This seems beneficial for LLMs in understanding video content.
2. The figures in the manuscript effectively illustrate the pipeline of the method and provide clear examples of entries in LangRepo, making it easy to follow.
3. Extensive experiments on various video question-answering datasets validate the impressive performance of LangRepo.

**Weaknesses:**

1. Table 1 shows that longer input captions can reduce LLM accuracy.  Adding details like frame rate, caption length, downsampling methods, and the effects of adjusting frame rates, rather than just subsampling or replicating captions, would make the findings more convincing.
2. The authors use CLIP to group redundant text. However, as seen in the example from Figure 4, some non-redundant sentences are quite similar to grouped ones, differing only in terms like "man X" and "Person C," which are semantically very similar, raising concerns about CLIP's ability to distinguish them. Additionally, it's unclear how CLIP's similarity threshold is set for grouping, as there are no ablation experiments shown. What's more, directly using LLMs for grouping might be more effective, and a quantitative comparison could be beneficial.
3. If a more effective VLLM or LLM is used, would LangRepo achieve consistent performance improvements? Specifically, the effectiveness of LangRepo with captions extracted by different VLLMs, and whether captions extracted by LangRepo perform better across various LLMs, requires further validation.

**Questions:**

See weakness above.

---

> ### Author Response · Authors · 2024-11-23
> **Response to reviewer VDGU [1/2]**
>
> **W1: Clarifications about Table 1: input length, caption frame-rate and caption sampling methods.**
>
> In Table 1, the 1x setting corresponds to the original captioning frame-rates (1fps). On average, this results in \~1.8k tokens of information per video (or, \~6 words/cap x 180 cap = \~1080 words/video) in EgoSchema, and \~0.9k tokens (or, \~14 words/cap x \~45 cap = \~630 words per video) in the other two datasets.
>
> By default, we create the 0.5x setting by subsampling the original captions and the 2x setting by resampling— decreasing/increasing the average token count accordingly (*e.g.* \~0.9k and \~3.6k tokens respectively in EgoSchema). Here, we sample each caption as-a-whole, rather than adjusting its word-length to avoid any changes to the semantics/meaning (*i.e.,* analogous to changing the captioning framerate). It allows us to keep both these settings as faithful as possible to the original 1x setting, and isolate the issue of LLM context utilization. We show the performance of these settings in table below. We will include these details in the paper.
>
> | Dataset | 0.5x || 1x || 2x ||
> |----------|----------|----------|----------|----------|----------|----------|
> | 		| #tokens 	| Acc % |  #tokens	| Acc % |  #tokens 	| Acc % |
> | EgoSchema 	| **~0.9k**	| 49.8 	|  **~1.8k**	| 48.8 	| **~3.6k** 	| 46.8 	|
> | NeXT-QA 	|**~0.45k**	| 48.2 	|  **~0.9k**	| 48.2 	| **~1.8k** 	| 46.9 	|
> | IntentQA  	|**~0.45k**  	| 47.1 	|  **~0.9**	| 46.9 	| **~1.8k** 	| 45.2 	|
>
> *all new numbers are in bold.*
>
>
> **W2.1: On the sensitivity of CLIP embeddings, for identifying fine-grained details in captions.**
>
> This is a valid concern by the reviewer. We agree that the similarity based grouping procedure can have noise, as we do this in zero-shot without have to retrain embedding modules. However, we empirically observe that this module based on pretrained image embeddings (*e.g.* CLIP), can work reasonably-well (as we show in Fig 4 and A.1, despite a few noisy instances). This noise level can be controlled by tuning the hyperparameter: reduction-rate (x)— that decides the percentage of caption similarities we group and rephrase. We empirically find x=0.25 to be a reasonable rate, that balances both extreme cases of (a) not grouping redundant captions, and (b) grouping non-redundant captions. We present a new ablation in this rebuttal to validate this claim (see below).
>
> | Reduction-rate (x) | EgoSchema Acc % |
> |----------|----------|
> | **0.10**	| **55.4**	|
> | 0.25 		|  57.8		|
> | **0.50**	| **56.2**	|
>
> *all new numbers are in bold.*
>
> Moreover, we can isolate the benefit of the redundancy-removal step by comparing chunk-based LLoVi (57.8%) vs. LangRepo (60.8%) in Table 6h. We also observe that CLIP-based embeddings (trained with category-based contrastive loss) outperforms Sentence-T5 (trained with sentence-level contrastive loss) on removing redundancies (57.8% vs. 56.4%). We will highlight this discussion in the final version of the paper.
>
>
> **W2.2: Can we directly-use an LLM to group+rephrase w/o relying on CLIP embeddings?**
>
> This is a good point. Following the suggestion from the reviewer, we consider this experiment of LLM-based grouping. However, we observe that the performance is significantly lower compared to our CLIP-based grouping (see table below).
>
> | Grouping 		| EgoSchema Acc %	|
> |----------|----------|
> | CLIP-based		| 57.8			|
> | **LLM-based**	| **52.6**		|
>
> *all new numbers are in bold.*
>
>
> When examining the rephrased descriptions closely, we identify a few potential reasons for this, which we will discuss in the final version of the paper.
>
> (1) In theLLM-based approach, it is hard to control which captions get grouped+rephrased. However, we rely on such structured information within our repository entries (*e.g.* which captions and how many are being grouped) to generate metadata that are useful for QA  (*e.g.* timestamps, occurrences).
>
> (2) Despite our best efforts with prompting, we are unable to reduce the noise in groupings in the LLM-based approach. With many captions as the input, the LLM tends to group+rephrase unrelated ones quite frequently, by simply combining them (*e.g.* with a comma).

---

> ### Author Response · Authors · 2024-11-23
> **Response to reviewer VDGU [2/2]**
>
> **W3: Performance of LangRepo with larger models.**
>
> We thank the reviewer for introducing this comparison, as it can reveal the scaling behavior of LangRepo.
>
> (1) When using a larger captioner to generate better captions as LangRepo input, the performance will be capped by the oracle performance (69.2% with LangRepo-7B) as we show in Table 6g.
>
> (2) When using a larger question-answering LLM, LangRepo scales its performance as we show in the new results that we report with this rebuttal (see table below). Here, we use LLama3.1-70b as the LLM, evaluating both LLoVi and LangRepo for a fair comparison. We see a consistent performance boost on both frameworks, while LangRepo still comfortably outperform LLoVi, both on EgoSchema subset (+4.8%) and fullset (+3.8%). We will discuss these results in the final version of the paper.
>
> | Model | Params | ES-Subset Acc % | ES-fullset Acc % |
> |----------|----------|----------|----------|
> | LLoVi 		| 7B 		| 50.8 		| 33.5 		|
> | LangRepo 	| 7B 		| 60.8 		| 38.9 		|
> | **LLoVi** 	| **70B** 	| **62.2** 	| **40.2** 	|
> | **LangRepo** | **70B** 	| **67.0** 	| **44.0** 	|
>
> *all new numbers are in bold.*

---

> ### Author Response · Authors · 2024-11-25
> **Follow-up**
>
> Dear Reviewer VDGU,
>
> Thank you again for your constructive feedback and time/effort reviewing our paper. Since the rebuttal period is ending soon, please let us know if our responses have addressed your concerns. We are happy to engage in further discussion to provide more clarifications if needed.
>
> Kind Regards!

---

> > ### Author Response · Authors · 2024-11-27
> > **Follow-up (2)**
> >
> > We thank the reviewer VDGU again for the initial feedback, which will definitely improve the quality and clarity of this paper. We believe we have addressed all of the reviewer's concerns, but we would be very happy to engage in further discussion and provide more clarifications if needed. Please let us know if the concerns have been addressed.
> >
> > Thanks so much!

---

> > > ### Author Response · Authors · 2024-12-02
> > > **Follow-up (3)**
> > >
> > > Dear Reviewer VDGU,
> > >
> > > Thank you again for your constructive feedback and time/effort reviewing our paper. Since the rebuttal period is ending soon, please let us know if our responses have addressed your concerns. We are happy to engage in further discussion to provide more clarifications if needed.
> > >
> > > We hope that our additional experiments and clarifications will be kindly considered in the final rating.
> > >
> > > Kind Regards!

---

> > > > ### Author Response · Authors · 2024-12-03
> > > > **Follow-up (4)**
> > > >
> > > > Dear Reviewer VDGU,
> > > >
> > > > We thank you again for your initial feedback. Please let us know if your concerns have been addressed. Since the rebuttal period is ending soon, we would really appreciate it if our additional experiments and clarifications can be considered in the final rating.
> > > >
> > > > Thanks so much!

---

### Official Review · Reviewer_NBPz · 2024-11-04

**Soundness:** 3
**Presentation:** 3
**Contribution:** 3
**Rating:** 6
**Confidence:** 4

**Summary:**

The paper focuses on the problem that LLM's ability of long-term information gradually declines with input length, which is an important and interesting problem to study. The authors propose a textual-only video representation approach to deal with the information decay problem. Specifically, the overall pipeline contains read and write operations. Writing operation describes the video content with texts and stores. Reading operation read the description and summarize the content for text QA to finish VQA in an all-textual approach. Though the all-textual video question answering has been explored in recent literatures, such Language-Repo approach remains interesting. The experiments show the superiority of the method on long-video benchmarks, such as EgoSchema, NExT-QA, IntentQA and NExT-GQA.

**Strengths:**

1. Using language as an intermediate modality to deal with long-term information decay in LLM makes sense and sounds interesting. This could be a popular trend and this work would play an important role. However, the authors should do a detailed literature review and thorough discussion of this trend.
2. The code and implementation of the methods is provided which will greatly ease the reproduction process.
3. While in a zero-shot setting and no video pretraining is required, the proposed method achieves promising results on all the involved long video benchmarks.

**Weaknesses:**

1. The idea of transforming visual content to texts and performing efficient question-answering has been explored in image [1] and video [2][3] domains. There should be a separate subsection in related work to discuss this trend and the difference, which is critical.
2. Reasonability of experiments in Table 1: the authors argue that by varying the input length into QA-LLM (such as Mistral-7B), it can derive the conclusion that longer text input produces worse performance. However, this requires several strict assumptions: first the captions in different lengths contain evenly distributed and all adequate video content information. If it is not, it could be the QA benchmark bias to derive the performance changes, that the answers are hind in the fixed part of the video (say the early part of the captions/videos); Secondly,  the input to Mistral-7B is long enough to trigger the information decay issue. I am doubting if the input is video caption only, the input context length might not be sufficient to discuss long-term information processing decay.
2. As illustrated in Figure 4, the group operation is not conducted in a strict temporal order (like t=2 is grouped with 26-28 and 37). Will there be a negative impact on temporal sensitive tasks and errors in describing action/event orders?
3. Experiments - Proved by SUM-shot[2], the all-textual approach should be performing well, as well as in short video understanding. Comparison should be included on MSRVTT-QA and ANet-QA (which is missing in current long video experiments).
4. Experiments - Extremely long videos: there have been several emerging long video benchmarks, such as Video-MME, Event-Bench and ShortFilms. It would be great and more promising to provide results on these benchmarks and do a comparison/discussion to discuss the potential of dealing with extremely long videos (which should work, given the motivation and design pipeline of LanguageRepo).
5. It would be interesting to explore how vision-language hallucination rate varies. Short video LLMs + Texts vs. Long video LLMs. Will the intermediate modality increase or decrease the hallucination? This is not mandatory but I think is interesting to explore.
6. Formulation in Eq10 should be improved. The use of $\texttt{vqa}$ here would be confusing.
7. Table 6 (g) shows the oracle captioner achieved 69.2 on EgoSchema sub-set. It means the performance upper bound of the method. It might not be a good signal for an extendable framework. Please make further discussion.
8. It is important to ensure the Eq10 process can parse the formated video content and won't lead to long-context information processing decay issues. Please make analysis and explainations.

[1] From images to textual prompts: Zero-shot visual question answering with frozen large language models
[2] Shot2Story20K: A New Benchmark for Comprehensive Understanding of Multi-shot Videos
[3] A simple llm framework for long-range video question-answering [already included in the paper]

**Questions:**

1. In experiment Table 1, what is the specific input of Mistral-7B, and how long is the input? It is important to show how to obtain 1X, 0.5X and 2X captions. What about prompting VLLM to generate descriptions in different lengths (complexity)?
2. How do you get the video chunks? Even sampling frames or via some semantic approach, such as shot detection. For each video clip, how to use image LLaVA to obtain a caption set for videos?
3. What if using powerful LLM as an oracle for QA, e.g., Gemini and GPT to replace the models in Table 6a.
4. The input description is highly formated and seems long and redundant (might be inevitable, since the video is split into chunks and does separate text summarization). How to ensure the LLM can parse such format and won't trigger the long-context information decay issue?

---

> ### Author Response · Authors · 2024-11-22
> **Response to reviewer NBPz [1/4]**
>
> **W3: Does the lack of temporal order in LangRepo, affect temporally-sensitive question-answering?**
>
> This is a valid concern. In LangRepo, we create an intermediate representation (*i.e.,* repository entries), which is not necessarily temporally-ordered. However, to avoid any loss of temporal information (*e.g.* order, locality), we preserve the corresponding timestamps as metadata within repository entries. This helps us correctly answer temporally-sensitive questions (as also discussed in L937-942 in supplementary). For instance, LangRepo outperforms LLoVi (which is *temporally-ordered*) on temporal-order questions in NExT-QA dataset (51.4% vs. 51.2%), and on Before/After questions in IntentQA dataset (47.8% vs. 45.1%), as reported in Table 3 and 4. We will expand this discussion in the final version of the paper.
>
>
>
> **W4, W5: LangRepo performance on short-video (*e.g.* MSRVTT-QA) and very-long video (*e.g.* Short Film Dataset) question-answering.**
>
> We thank the reviewer for bringing these benchmark comparisons to our attention. We agree that these two extremes (short-video and very long-video) provide meaningful context to the reader in terms of the generalization of LangRepo. Hence, we select two representative benchmarks (MSRVTT-QA for short-video and ShortFilms for very long-video) and report comparisons, in addition to the already-reported 4 long-video benchmarks in the main paper.
>
> | Model 		| MSRVTT-QA Acc % (confidence) |
> |----------|----------|
> | VideoChat	| 45.0 (2.5)	|
> | MovieChat	| 52.7 (2.6)	|
> | VideoChat2	| 54.1 (3.3)	|
> | IG-VLM	| 63.7 (3.5)	|
> | **LLoVi**	| **58.6 (2.9)**	|
> | **LangRepo**	| **59.2 (3.0)**	|
>
> *all new numbers are in bold.*
>
> Here, we compare the performance of LangRepo with similar-sized models (7B) on MSRVTT-QA. We rely on LLaVA-1.5 captions (at 4fps) to answer the open-ended questions, which are then evaluated with an *LLM-as-a-judge* that predicts the accuracy (and confidence-score). We see that LangRepo achieves a competitive performance compared to other baselines, while outperforming LLoVi at the same experimental setup. This validates the usefulness of LangRepo even in short-video QA.
>
> | Model | Temporal window | SFD Acc % (change) |
> |----------|----------|----------|
> | FrozenBiLM	| Scene-level	| 22.7		|
> | FrozenBiLM	| Movie-level	| 23.4 (+0.7)	|
> | LLoVi		| Scene-level	| 34.2		|
> | LLoVi		| Movie-level	| 30.8 (-3.4)	|
> | LangRepo	| Scene-level	| 26.9		|
> | LangRepo	| Movie-level	| 29.0 (+2.1)	|
>
> On Short Film Dataset (SFD), the authors of the dataset paper already included LangRepo as one of their standard baselines for very long-video QA. This fact validates that LangRepo is being already adopted in the community. In SFD, various temporal window-sizes (shot-level -> scene-level -> movie-level) are considered for evaluation— where movie-level averages ~13mins. Here, we present an interesting observation when going from scene-level to movie-level (containing all scenes), that corresponds to very long-video QA with *visual information given as text*— the closest setup to our original LangRepo evaluation.
>
> Even without any hyperparameter tuning, LangRepo performs competitively with similar baselines (FrozenBiLM, LLoVi), while also showing a striking difference when going up to movie-level (+2.1%), compared to LLoVi (-3.4) and FrozenBiLM (+0.7). This shows third-party evidence that LangRepo can generalize to very long-videos.
>
> We will include both these extremes and the corresponding discussions in the final version of the paper.
>
>
> **W8: Does the oracle performance in Table 6g, show the absolute upper-bound of LangRepo?**
>
> This is a valid concern. We want to highlight that the oracle (in Table 6g) shows the performance upper-bound **w.r.t. the captioner (or, the input quality)**. This is a pre-processing step that converts all visual information to text, and although LangRepo relies purely on such information, this is not the absolute performance upper-bound of LangRepo.
> (1) In this paper, we experiment with 7B/12B scale open-source models (having better accessibility), but this can be scaled to a much-larger size (*e.g.* 1.8T in GPT-4). As seen with the scaling behavior of other text-only models (*e.g.* LLoVi), we expect the performance to scale with model size.
>
> (2) LangRepo relies on VLM embeddings such as the ones from CLIP/Sentence-T5 to identify redundancies. With more-recent and better visual embedders, it can identify redundancies more-accurately, resulting in higher reduction rates (*i.e.,* better context-utilization) and better performance. Being a generic *zero-shot* framework, LangRepo enables easier modularization and extendability of these components.
>
> (3) With more compute availability, we can afford to make compute-heavy decisions (*e.g.* higher number of temporal-scales, repository entries) to achieve a better performance.
>
> We will clarify and include this discussion in the final version of the paper.

---

> ### Author Response · Authors · 2024-11-22
> **Response to reviewer NBPz [2/4]**
>
> **Q1.1: Clarifications about Table 1: input length and caption sampling methods.**
>
> In Table 1, the 1x setting corresponds to the original captioning frame-rates (1fps). On average, this results in \~1.8k tokens of information per video in EgoSchema, and \~0.9k tokens in the other two datasets. By default, we create the 0.5x setting by uniformly-subsampling the original captions and the 2x setting by uniformly-resampling— decreasing/increasing the average token count accordingly (*e.g.* \~0.9k and \~3.6k tokens respectively in EgoSchema). It allows us to keep both these settings as faithful as possible to the original 1x setting, and isolate the issue of LLM context utilization. We show the performance of these settings in table below. We will include these details in the paper.
>
> | Dataset | 0.5x || 1x || 2x ||
> |----------|----------|----------|----------|----------|----------|----------|
> | 		| #tokens 	| Acc % |  #tokens	| Acc % |  #tokens 	| Acc % |
> | EgoSchema 	| **~0.9k**	| 49.8 	|  **~1.8k**	| 48.8 	| **~3.6k** 	| 46.8 	|
> | NeXT-QA 	|**~0.45k**	| 48.2 	|  **~0.9k**	| 48.2 	| **~1.8k** 	| 46.9 	|
> | IntentQA  	|**~0.45k**  	| 47.1 	|  **~0.9**	| 46.9 	| **~1.8k** 	| 45.2 	|
>
> *all new numbers are in bold.*
>
>
> **Q1.2: Thoughts on other methods of obtaining variable-length captions in Table 1 (*e.g.* prompting VLLM, rephrasing with LLM).**
>
> This is an interesting extension. First, we want to note that it is challenging to control the settings to be exactly 0.5x/2x with a VLLM directly, due to their limited prompt following (*e.g.* LLaVA-1.5, in contrast to premier LLMs)— to generate captions with an exact word length. Moreover, this may not be an ideal setting to isolate the issue of context utilization, as such captions bring varying levels of information (or, hallucination).
>
>
> Therefore, in this rebuttal, we try LLM-based rephrasing (*e.g.* with LLama3.1-70B) to create alternate versions of EgoSchema 0.5x and 2x captions. We do not change the caption structure (i.e., subject or object), keeping them faithful to the 1x setting. We consider two types of prompts, either (1) to change caption length keeping the number of captions constant, or (2) change the number of captions (as we did with our original subsampling/resampling, but this time with an LLM).  However, despite our best effort, both these strategies does not work as expected. The reasoning is as follows.
>
> In (1), it is not possible to reduce the caption length in-half as the captions are already of a minimal length already (*e.g.* `#C C picks phone.`)— as is the case with oracle captions in EgoSchema. If we ask to increase the length, it usually hallucinate details (*e.g.* `#C C picks phone to call someone.`). Therefore, the strategy in (1) is not giving faithful 0.5x, 2x settings.
>
> The above (2) is basically not that different from our original setup of subsampling/resampling. The LLM output (corresponding to 0.5x setting) usually starts with `I'll select every other sentence to generate a smaller description …`, and the selected captions are in-fact almost the same as the original subsampled captions (w/ very minimal noise). Therefore, we do not expect the performance to change from our original setting.
>
> We will include this discussion in the supplementary.
>
>
> **Q3: Performance of more-powerful LLMs in Table 6a ablation.**
>
> This is an interesting ablation. Following the suggestion from the reviewer, we run a new baseline in this rebuttal with LLama-3.1-70b open-source LLM, which is shown to have a comparable performance with GPT-3.5 scale models. The observations are given in the table below. It shows that the performance of this fully text-based video VQA approaches (*e.g.* LLoVi, LangRepo) can be scaled with model size.
>
> | Question-answering LLM | Params. | EgoSchema Acc % |
> |----------|----------|----------|
> | LLama2	| 13B		| 43.0		|
> | Mistral	 	| 7B		| 50.8		|
> | **Llama3.1**	| **70B**	| **62.2**	|
>
> *all new numbers are in bold.*

---

> ### Author Response · Authors · 2024-11-22
> **Response to reviewer NBPz [3/4]**
>
> **W2.1: Does the claim in Table 1 rely on assumption “information is uniformly distributed”? Or, may the observation be due to a dataset bias?**
>
> This is a valid concern. First, let us reiterate what we observe in Table 1, and what we hypothesize based on that. We observe that *the VQA performance drops with increasing number of captions*. Based on this observation, we hypothesize that *this phenomena is due to the increased context utilization of the question-answering LLM*.
>
> Here, our experiment does not rely on any assumption on information distribution. Regardless of uniformly or non-uniformly distributed information in input captions, the observation holds. Any information that is in 1x setting, is available at 2x, and if the LLM does not have a context utilization issue, the performance should at least remain constant when going from 1x to 2x. This behavior is also observed in literature, even with much-larger LLMs (*e.g.* Gemini), and studied in-detail in a concurrent work [1].
>
> Moreover, this observation may not likely be due to a dataset bias, as we experiment with 3 prominent long-video VQA benchmarks in Table 1 (*e.g.* EgoSchema, NExT-QA, IntentQA), and observe the same behavior. We will better clarify this reasoning in the final version of the paper.
>
>
> **W2.2: Does the claim in Table 1 rely on assumption “input length is sufficient to trigger information decay”?**
>
> We agree that our claim relies on this assumption, which is supported by what we observe in Table 1. Yet, we would like to clarify what we mean by *’information decay’* here. It does not mean a *hard-trigger* such as context overflow/truncation, but rather a *soft-trigger* of losing effectiveness of information processing.
>
> Let us put things into perspective more-concretely. Here, we use Mistral-7B, which has a context-length of 8k tokens. The input captions of all datasets and settings in Table 1, can fit well-within this context length (*e.g.* EgoSchema 2x has \~3.6k tokens, and NExT-QA/IntentQA 2x has \~1.8k tokens, as shown in the table for Q5). This means, we will not be hard-triggering a context overflow (even after considering prompt tokens), but rather the observation is due to a soft-trigger of information decay.  This behavior is shown to be true even for much-larger LLMs such as Gemini [1]. It is attributed to the attention mechanism being overwhelmed with increasingly-longer inputs. We will highlight these details to clarify/strengthen our claim in Table 1.
>
>
> **W9: Can LangRepo descriptions be processed by the question-answering LLM w/o information decay?**
>
> This is a valid concern. The descriptions from LangRepo are less-likely to trigger an information decay, compared to naive captions (as shown in Table 6i). Let us validate this claim more-concretely, by looking into LLM context-lengths and input token-lengths.
>
> The two main LLMs that we use— Mistral-7B or Mixtral8x7B— have 8k or 32k context-lengths. However, such LLMs lose their effectiveness well-below this context limit, as the attention mechanism gets overwhelmed with increasingly-longer inputs— which we identify as *information decay* (in Table 1), rather than the *context truncation/overflow*. This is true even for much-larger LLMs such as Gemini, as shown in [1].
>
> In our EgoSchema setup, LangRepo generates \~0.7k tokens for a 3-min video (compared to \~1.8k tokens in naive captions)— still comfortably fitting into the context of these models. Even for longer videos (\~13min) such as in SFD benchmark, the context utilization will be \~3.5k with a similar pipeline (compared to \~9k tokens with naive captions). This means LangRepo will usually be less-prone to information decay. However, the context-utilization of LangRepo depends on its hyperparameters (*e.g.* number of temporal-scales, summarization length, captioning frame-rate), and should be taken into account when adopting it to newer settings. We will include this discussion in the supplementary.
>
> [1] Same Task, More Tokens: the Impact of Input Length on the Reasoning Performance of Large Language Models [arXiv 2024]

---

> ### Author Response · Authors · 2024-11-22
> **Response to reviewer NBPz [4/4]**
>
> **W1: Include a related work subsection on text-only VQA models.**
>
> We thank the reviewer for raising this important discussion. We agree that expanding our current discussion (which cites [3,5,7,8]) will provide a better context to the reader.
>
> There exists a body of research that relies purely on text information to perform visual question-answering (eg: [1,4] for image-VQA, [8,9,10] for video-VQA). These pipelines usually include two-stages: (1) converting visual-information to text and (2) question-answering, and the corresponding literature either focus on stage-1 [2,5,6] or stage-2 [3,7] above. Our Language Repository (LangRepo) focuses on stage-2. Closest to ours are LLoVi [3] (in-terms of its summarization), Video ReCap [6] (in-terms of its multi-scale descriptions), and MVU [11] (in-terms of its multi-modal information as text)— some of which are concurrent work [6,11].
> Different from these, our novelty is on *removing redundancies in visual information given as text* across varying scales (based on iterative refinement), addressing the problem of effective context utilization of question-answering LLMs. We will better highlight this in the final version of the paper.
>
>
> **W6: [Optional] Thoughts on hallucination in (a) Long-video MLMs vs. (b) Short-video Captioner + LLMs (*e.g.* LangRepo setup)**
>
> We agree that this would be an interesting discussion. One hypothesis based on our observations is as follows. Short-video Captioners usually provide more-precise initial captions, which can effectively be aggregated by a second-stage LLM (as also shown in LLoVi). However, since the captioner lack long-range temporal context, the generated captions can be *highly-redundant*, rather than being *hallucinative*. In contrast, a Long-video multi-modal model (MLM) usually consumes a large number of tokens at once, where the long-context attention mechanism becomes more prone to hallucination. We leave a more-systemic evaluation of this phenomenon to future work in the direction.
>
>
> **W7: Making Eq. 10 in the paper more-precise.**
>
> We thank the reviewer for raising this concern. We agree that since all visual information is already converted to text, using `vqa(.)` in Eq. 10 can be misleading. We will rename this more-appropriately: *e.g.* `generate-answer(.)` or `question-answering(.)`.
>
>
> **Q2.1: Details about video chunking (uniform or shot-detection based).**
>
> To create chunks within the LangRepo framework, we rely on uniform chunking (w/o any shot detection that adds an extra overhead and added advantage). For instance, if we want to have 4-chunks in a 180-second video (given as 180 captions), we group 45 captions each as non-overlapping chunks. We will better clarify this in the final version of the paper.
>
> **Q2.2: Details about caption generation .**
>
> We use the same initial captions generated by the authors of LLoVi paper, for fair comparison. For instance, on EgoSchema we have captions extracted at 1fps. When using LLaVA (image-based VLM) for captioning, a single frame is sampled at 1fps and captioned. When using LaViLa (video-based VLM) for captioning, a very-short clip is sampled within each temporal window at 1fps and captioned. We will clarify these details in the supplementary.
>
> [1] From images to textual prompts: Zero-shot visual question answering with frozen large language models [CVPR 2023]
>
> [2] Shot2Story20K: A New Benchmark for Comprehensive Understanding of Multi-shot Videos [arXiv 2023]
>
> [3] A simple LLM framework for long-range video question-answering [EMNLP 2024]
>
> [4] An Empirical Study of GPT-3 for Few-Shot Knowledge-Based VQA [AAAI 2022]
>
> [5] Learning video representations from large language models [CVPR 2023]
>
> [6] Video ReCap: Recursive Captioning of Hour-Long Videos [CVPR 2024]
>
> [7] Socratic Models: Composing Zero-Shot Multimodal Reasoning with Language [ICLR 2023]
>
> [8] VideoChat : Chat-Centric Video Understanding [CVPR 2024]
>
> [9] Video ChatCaptioner: Towards Enriched Spatiotemporal Descriptions [arXiv 2023]
>
> [10] ChatVideo: A Tracklet-centric Multimodal and Versatile Video Understanding System [arXiv 2023]
>
> [11] Understanding Long Videos with Multimodal Language Models [arXiv 2024]

---

> ### Author Response · Authors · 2024-11-25
> **Follow-up**
>
> Dear Reviewer NBPz,
>
> Thank you again for your constructive feedback and time/effort reviewing our paper. Since the rebuttal period is ending soon, please let us know if our responses have addressed your concerns. We are happy to engage in further discussion to provide more clarifications if needed.
>
> Kind Regards!

---

> > ### Author Response · Authors · 2024-11-29
> > **Follow-up (2)**
> >
> > We thank the reviewer NBPz again for the initial feedback, which will definitely improve the quality and clarity of this paper. We believe we have addressed all of the reviewer's concerns, but we would be very happy to engage in further discussion and provide more clarifications if needed. Please let us know if the concerns have been addressed.
> >
> > Thanks so much!

---

> > > ### Author Response · Authors · 2024-12-03
> > > **Follow-up (3)**
> > >
> > > Dear Reviewer NBPz,
> > >
> > > Thank you again for your constructive feedback and time/effort reviewing our paper. Since the rebuttal period is ending soon, please let us know if our responses have addressed your concerns. We are happy to engage in further discussion to provide more clarifications if needed.
> > >
> > > We hope that our additional experiments and clarifications will be kindly considered in the final rating.
> > >
> > > Kind Regards!

---

### Official Review · Reviewer_5tXY · 2024-11-20

**Soundness:** 3
**Presentation:** 3
**Contribution:** 2
**Rating:** 8
**Confidence:** 3

**Summary:**

This paper introduces a video modeling method called LangRepo, which forms a text data organization strategy. The method performs well on four datasets, demonstrating its effectiveness.

**Strengths:**

1. The newly proposed concept of using concise text as a representation for video reveals why current captioning-based long video understanding methods perform well, which is enlightening.
2. The authors conducted observations and various ablation experiments to demonstrate the effectiveness of their approach.
3. Experiments on multiple datasets show consistent improvements compared to the most similar related work, LLoVi.

**Weaknesses:**

1. The usage of log-likelihood is not fair, as other models do not utilize it, especially on the IntentQA dataset.
2. The experiments on long videos are not sufficient. NeXT-QA and NeXT-GQA are not typical long video benchmarks. At least the long split in VideoMME[1] should be incorporated. Otherwise, the authors should not stress long videos in the title, since textual representation for videos is also valuable for short videos.
3. The observation experiment in Table 1 is hard to interpret; it would be helpful if the token sequence length were provided.


[1] Video-MME: The First-Ever Comprehensive Evaluation Benchmark of Multi-modal LLMs in Video Analysis.

**Questions:**

N/A

---

> ### Author Response · Authors · 2024-11-23
> **Response to reviewer 5tXY [1/1]**
>
> **W1: Do main results tables make a fair comparison (w/ Log-likelihood classifier)?**
>
>
> We sincerely apologize for the confusion. We use the LL-classifier in *all* our main benchmark experiments (*e.g.* EgoSchema, NExT-QA, IntentQA, NExT-GQA), as we consider it to be an inclusive component of the LangRepo framework. Such usage is also discussed in the supplementary  (L983-1003), when we introduce specific prompting styles with the LL-classifier for each underlying dataset.
>
> To make sure we still do a fair comparison, the results corresponding to the closely-related baselines (*e.g.* Mistral, LLoVi) that we already include in our main tables (Table 2-5), use the same LL-classifier setup (instead of Gen-classifier). However, replicating the results for all other baselines under LL-classifier is prohibitively-expensive. For better clarity, we additionally provide the performance on EgoSchema with the two different classifiers in the table below, extending the results in Table 2 and 6c. We also note that other concurrent work (*e.g.* TOPA [arXiv 2024], MVU [arXiv 2024]) also adopt LL-classifier for MCQ benchmarks. We will better clarify the evaluation setting for each experiment in the final version of the paper, avoiding such confusion.
>
>
> | Model 			| w/ LL-classifier		| w/ Gen-classifier |
> |----------|----------|----------|
> | Mistral (7B) 		| 48.8 			| **47.2**	|
> | LLoVi (7B) 		| 50.8 			| **50.2** 	|
> | LangRepo (7B)	| 60.8 			| 58.8 		|
>
> *all new numbers are in bold.*
>
>
> **W2: LangRepo performance on very-long video (*e.g.* Short Film Dataset) question-answering.**
>
> We thank the reviewer for bringing this benchmark comparisons to our attention. We agree that evaluations on very long-video QA show the generalization of LangRepo. Hence, in this rebuttal, we include results on ShortFilms Dataset (average length of ~13mins).
>
> | Model | Temporal window | SFD Acc % (change) |
> |----------|----------|----------|
> | FrozenBiLM	| Scene-level	| 22.7		|
> | FrozenBiLM	| Movie-level	| 23.4 (+0.7)	|
> | LLoVi		| Scene-level	| 34.2		|
> | LLoVi		| Movie-level	| 30.8 (-3.4)	|
> | LangRepo	| Scene-level	| 26.9		|
> | LangRepo	| Movie-level	| 29.0 (+2.1)	|
>
> Here, the authors of the dataset paper already included LangRepo as one of their standard baselines for very long-video QA. This fact validates that LangRepo is being already adopted in the community. In SFD, various temporal window-sizes (shot-level -> scene-level -> movie-level) are considered for evaluation. We present an interesting observation (see the table above), when going from scene-level to movie-level, that corresponds to very long-video QA with *visual information given as text*— the closest setup to our original LangRepo evaluation.
>
> Even without any hyperparameter tuning, LangRepo performs competitively with similar baselines (FrozenBiLM, LLoVi), while also showing a striking difference when going up to movie-level (+2.1%), compared to LLoVi (-3.4) and FrozenBiLM (+0.7). This shows third-party evidence that LangRepo can generalize to very long-videos. We will include this discussion in the final version of the paper.
>
>
>
> **W3: Clarifications about Table 1: input length, caption frame-rate and caption sampling methods.**
>
> We sincerely apologize for the lack of information. In Table 1, the 1x setting corresponds to the original captioning frame-rates (1fps). On average, this results in \~1.8k tokens of information per video (or, \~6 words/cap x 180 cap = \~1080 words/video) in EgoSchema, and \~0.9k tokens (or, \~14 words/cap x \~45 cap = \~630 words per video) in the other two datasets.
>
> By default, we create the 0.5x setting by subsampling the original captions and the 2x setting by resampling— decreasing/increasing the average token count accordingly (*e.g.* \~0.9k and \~3.6k tokens respectively in EgoSchema). Here, we sample each caption as-a-whole, rather than adjusting its word-length to avoid any changes to the semantics/meaning (*i.e.,* analogous to changing the captioning framerate). It allows us to keep both these settings as faithful as possible to the original 1x setting, and isolate the issue of LLM context utilization. We show the performance of these settings in table below. We will include these details in the paper.
>
> | Dataset | 0.5x || 1x || 2x ||
> |----------|----------|----------|----------|----------|----------|----------|
> | 		| #tokens 	| Acc % |  #tokens	| Acc % |  #tokens 	| Acc % |
> | EgoSchema 	| **~0.9k**	| 49.8 	|  **~1.8k**	| 48.8 	| **~3.6k** 	| 46.8 	|
> | NeXT-QA 	|**~0.45k**	| 48.2 	|  **~0.9k**	| 48.2 	| **~1.8k** 	| 46.9 	|
> | IntentQA  	|**~0.45k**  	| 47.1 	|  **~0.9**	| 46.9 	| **~1.8k** 	| 45.2 	|
>
> *all new numbers are in bold.*

---

> > ### Author Response · Authors · 2024-11-25
> > **Follow-up**
> >
> > Dear Reviewer 5tXY,
> >
> > Thank you again for your constructive feedback and time/effort reviewing our paper. Since the rebuttal period is ending soon, please let us know if our responses have addressed your concerns. We are happy to engage in further discussion to provide more clarifications if needed.
> >
> > Kind Regards!

---

> > > ### Author Response · Authors · 2024-11-29
> > > **Follow-up (2)**
> > >
> > > We thank the reviewer 5tXY again for the initial feedback, which will definitely improve the quality and clarity of this paper. We believe we have addressed all of the reviewer's concerns, but we would be very happy to engage in further discussion and provide more clarifications if needed. Please let us know if the concerns have been addressed.
> > >
> > > Thanks so much!

---

> > > > ### Comment · Reviewer_5tXY · 2024-11-30
> > > > **Thank you for the response**
> > > >
> > > > Thank you for your response.
> > > >
> > > > I believe the authors have effectively addressed my concerns regarding the methodologies and experiments. Additionally, this paper achieves superior results with fewer LLM parameters compared to other agent-based video understanding methods on the EgoSchema dataset. Consequently, I am raising my score to accept. However, I cannot assign a higher score because the paper's performance on other datasets (NExT-QA, IntentQA, and NExT-GQA) does not match its success on the EgoSchema dataset, suggesting potential generalization issues. I hope the authors will explore this aspect further in their future work.

---

### Author Response · Authors · 2024-11-22
**General Comment**

We thank all the reviewers for their constructive feedback and appreciate their time/effort reviewing our paper. In this rebuttal, we provide clarifications with evidence to answer reviewer concerns, as individual responses to each reviewer. Please let us know if further clarifications are needed during the rebuttal period.

---

### Meta-Review · Area_Chair_GWXK · 2024-12-18

**Metareview:**

This submission received two negative sores and two positive scores after rebuttal. The reviewer who assigned the score 8 still has a  potential generalization issue: paper's performance on other datasets (NExT-QA, IntentQA, and NExT-GQA) does not match its success on the EgoSchema dataset. After carefully reading the paper, the review comments, the AC can not recommend the acceptance of this submission, as the average score is under the threshold bar and the concerns about efficiency and effectiveness of proposed approach remain. The AC also recognizes the contributions confirmed by the reviewers, and encourages the authors to update the paper according to the discussion and submit it to the upcoming conference.

**Additional Comments On Reviewer Discussion:**

After discussion, while reviewer # NBPz and #5tXYthought that the responses fully addressed their concerns reviewer#Mbnb still has major concerns about the experimental validation of the proposed approach and keeps a negative score.

---

### Decision · Program_Chairs · 2025-01-22

Reject